# Munc18-1 catalyzes neuronal SNARE assembly by templating SNARE association

Junyi Jiao[1], Mengze He[1], Sarah A Port[2], Richard W Baker[2,3], Yonggang Xu[1], Hong Qu[1], Yujian Xiong[1], Yukun Wang[1], Huaizhou Jin[1], Travis J Eisemann[2], Frederick M Hughson[2]*, Yongli Zhang[1]*

[1]Department of Cell Biology, Yale University School of Medicine, New Haven, United States; [2]Department of Molecular Biology, Princeton University, Princeton, United States; [3]Department of Cellular and Molecular Medicine, University of California, San Diego, San Diego, United States

**Abstract** Sec1/Munc18-family (SM) proteins are required for SNARE-mediated membrane fusion, but their mechanism(s) of action remain controversial. Using single-molecule force spectroscopy, we found that the SM protein Munc18-1 catalyzes step-wise zippering of three synaptic SNAREs (syntaxin, VAMP2, and SNAP-25) into a four-helix bundle. Catalysis requires formation of an intermediate template complex in which Munc18-1 juxtaposes the N-terminal regions of the SNARE motifs of syntaxin and VAMP2, while keeping their C-terminal regions separated. SNAP-25 binds the templated SNAREs to induce full SNARE zippering. Munc18-1 mutations modulate the stability of the template complex in a manner consistent with their effects on membrane fusion, indicating that chaperoned SNARE assembly is essential for exocytosis. Two other SM proteins, Munc18-3 and Vps33, similarly chaperone SNARE assembly via a template complex, suggesting that SM protein mechanism is conserved.
DOI: https://doi.org/10.7554/eLife.41771.001

*For correspondence:
hughson@princeton.edu (FMH);
yongli.zhang@yale.edu (YZ)

Competing interests: The authors declare that no competing interests exist.

## Introduction

Cytosolic Sec1/Munc18 (SM) proteins and membrane-anchored SNARE proteins constitute the core machinery that mediates nearly all intracellular membrane fusion (*Rizo and Südhof, 2012*; *Südhof and Rothman, 2009*). In particular, the neuronal SM protein Munc18-1 and its cognate SNAREs syntaxin-1, SNAP-25, and VAMP2 (also called synaptobrevin) drive fusion of synaptic vesicles with the presynaptic plasma membrane (*Söllner et al., 1993*; *Verhage et al., 2000*). Fusion releases neurotransmitters into synaptic or neuromuscular junctions, controlling all thoughts and actions. Related SM proteins, Munc18-2 and Munc18-3, are required for cytotoxin release from lymphocytes to kill cancerous or infected cells (*Côte et al., 2009*) and for glucose uptake (*Bryant and Gould, 2011*), respectively. Consequently, dysfunctions of SM proteins are associated with neurological and immunological disorders, cancers, diabetes, and other diseases (*Bryant and Gould, 2011*; *Côte et al., 2009*; *Stamberger et al., 2016*).

SM proteins regulate the assembly of SNAREs into the membrane-bridging 'trans-SNARE' complexes required for membrane fusion (*Figure 1*) (*Baker and Hughson, 2016*; *Brunger et al., 2018*; *Gao et al., 2012*; *Rizo and Südhof, 2012*; *Shen et al., 2007*; *Südhof and Rothman, 2009*; *Sutton et al., 1998*). Most SNAREs contain a C-terminal transmembrane anchor, an adjacent SNARE motif, and an N-terminal regulatory domain (NRD). SNARE motifs are 60–70 residues in length, with either glutamine (Q-SNAREs) or arginine (R-SNAREs) residues at a key central position (*Fasshauer et al., 1998*). SNARE motifs in isolation are intrinsically disordered. By contrast, they are

**Figure 1.** Two potential pathways for Munc18-1-regulated neuronal SNARE assembly. (i) Munc18-1 first serves as a syntaxin chaperone and binds syntaxin to inhibit its association with other SNAREs. (ii) Closed syntaxin is opened by Munc13-1, a large multifunctional protein that also helps tether vesicles to the plasma membrane and binds, albeit with low affinity, both syntaxin and VAMP2. (iii) Open syntaxin may bind SNAP-25 to form a syntaxin:SNAP-25 or a Munc18-1:syntaxin:SNAP-25 complex. (iv) Alternatively, open syntaxin may bind VAMP2 to form a Munc18-1:syntaxin:VAMP2 template complex, as proposed here. Both complexes, (iii) and (iv), have been proposed to be 'activated' for SNARE assembly. (v) and (vi) Other factors such as synaptotagmin (not shown) target the half-zippered SNARE complex to enable calcium-triggered further SNARE zippering and vesicle fusion.

DOI: https://doi.org/10.7554/eLife.41771.002

α-helical in fusion-competent SNARE complexes, with three Q-SNARE motifs (designated Qa, Qb, and Qc) and one R-SNARE motif combining to form a parallel four-helix bundle (*Sutton et al., 1998*). Despite their apparent simplicity, however, the physiological pathway(s) of SNARE assembly have remained enigmatic, as have the specific role(s) of SM proteins (*Baker et al., 2015*; *Jakhanwal et al., 2017*; *Lai et al., 2017*; *Ma et al., 2013*; *Ma et al., 2015*; *Rizo and Südhof, 2012*; *Shen et al., 2007*; *Wickner, 2010*; *Zhang et al., 2015*; *Zhou et al., 2013*).

SNARE assembly has long been thought to begin with the formation of a t-SNARE complex among the SNAREs – usually Qa, Qb, and Qc – residing on the target membrane (*Weber et al., 1998*) (*Figure 1*). According to this view, the neuronal SNAREs syntaxin (Qa-SNARE) and SNAP-25 (Qbc-SNARE, a single protein containing both Qb and Qc SNARE motifs) assemble on the presynaptic plasma membrane, forming a t-SNARE complex that subsequently binds to the synaptic vesicle R-SNARE VAMP2 (*Jakhanwal et al., 2017*; *Pobbati et al., 2006*; *Shen et al., 2007*; *Weber et al., 1998*; *Zhang et al., 2016b*). Recent reports have, however, raised doubts about this order of events. In vitro reconstitution experiments suggested that neuronal SNARE assembly begins with a complex between Munc18-1 and syntaxin, requires Munc13-1, and may not involve a syntaxin:SNAP-25 complex (*Ma et al., 2013*) (*Figure 1*). Crystal structures of the SM protein Vps33 bound to its cognate Qa- and R-SNARE implied that the SM protein functions as a template, orienting and aligning the two SNARE motifs for further assembly (*Baker et al., 2015*). Thus, the Qa- and R-SNAREs might be the first to assemble, and only on the surface of an SM template.

Previously, we developed a single-molecule approach based on optical tweezers to dissect SNARE assembly at high spatiotemporal resolution (*Gao et al., 2012*; *Ma et al., 2015*; *Zhang et al., 2016b*; *Zorman et al., 2014*). Using this method, we measured the folding energy and kinetics of various SNARE complexes. Here, we extend the method to observe SM-mediated SNARE assembly. We detected three template complexes, each of them comprising an SM protein (Munc18-1, Munc18-3, or Vps33) bound to its cognate Qa- and R-SNAREs, and characterized the neuronal template complex in detail using a large panel of mutant proteins. Our results imply that the neuronal template complex is an on-pathway, rate-limiting intermediate in vitro and in vivo. They further suggest that phosphorylation of Munc18-1 can modulate the efficiency of neurotransmitter release by affecting the stability of the template complex. More broadly, our findings imply that membrane fusion in vivo may be controlled by SM proteins through their tunable catalytic activity as SNARE assembly chaperones.

## Results

### Munc18-1, syntaxin, and VAMP2 form a template complex

Previously, we found that the SM protein Vps33 forms binary complexes with the SNARE motifs of Vam3 (Qa-SNARE) and Nyv1 (R-SNARE), as well as a ternary complex containing all three proteins (*Baker et al., 2015*). Crystal structures of the two binary complexes revealed that the Qa-SNARE and the R-SNARE bind to adjacent sites on the SM protein and led to a model of the template complex in which the two SNARE motifs are 'half-zippered'. An analogous template complex might form during the assembly of the neurotransmitter release machinery (*Sitarska et al., 2017*), but direct evidence is lacking. To investigate further, we used an optical tweezers-based strategy to directly observe neuronal SNARE assembly and disassembly in the presence of Munc18-1. To mimic a trans-SNARE complex, pre-assembled SNAREs were attached via the C termini of the Qa- and R-SNARE motifs to beads (*Gao et al., 2012*). The same two SNARE motifs were covalently linked near their N termini through an engineered disulfide bond to form a Qa-R-SNARE conjugate (*Figure 2A* and *Figure 2—figure supplement 1*). This tactic permitted us to conduct repeated rounds of force-induced unfolding/disassembly ('pulling') and potential refolding/assembly ('relaxation') in a single experiment.

Munc18-1 binds both the Qa-SNARE syntaxin (with nanomolar affinity) and the R-SNARE VAMP2 (with micromolar affinity) (*Burkhardt et al., 2008*; *Misura et al., 2000*; *Parisotto et al., 2014*; *Sitarska et al., 2017*). Formation of a ternary template complex has not, however, been reported. This is presumably because Munc18-1 and syntaxin, in their high-affinity complex, both adopt conformations that preclude VAMP2 binding (*Baker et al., 2015*; *Misura et al., 2000*; *Sitarska et al., 2017*) (*Figure 2B*). In particular, the SNARE motif and NRD of syntaxin interact to create an autoinhibited or 'closed' conformation (*Misura et al., 2000*). Opening syntaxin, and thereby permitting SNARE assembly, requires Munc13-1 (*Ma et al., 2011*; *Ma et al., 2013*; *Wang et al., 2017*; *Yang et al., 2015*). To bypass the requirement for Munc13-1 in our single-molecule experiments, we attempted to destabilize the closed conformation of syntaxin without abolishing its interactions with Munc18-1 or the other SNAREs. Among the strategies we evaluated, the simplest was to form the Qa-R-SNARE conjugate by crosslinking syntaxin R198C and VAMP2 N29C (*Figure 2A*, solid arrowhead; *Figure 2—figure supplement 1*). In closed syntaxin, residue 198 is buried against the NRD (*Figure 2B*). As shown below, involving this residue in a disulfide bond destabilized and partially opened Munc18-bound syntaxin, presumably via localized unfolding.

We began by pulling the fully folded neuronal SNARE complex, containing crosslinked syntaxin and VAMP2 as well as SNAP-25B, in the absence of Munc18-1. The resulting force-extension curve (FEC) revealed that, as expected based on our previous work (*Gao et al., 2012*; *Ma et al., 2015*), the SNARE complex disassembled in at least three steps (*Figure 2C*, FEC #1, gray curve). These force-induced disassembly steps are schematically depicted in *Video 1* and in *Figure 2D* as transitions from states 1↔2→3→4. 1↔2 represents reversible unfolding of the C-terminal half of the VAMP2 SNARE motif (CTD; *Figure 2C*, gray oval), 2→3 represents irreversible unfolding of the N-terminal half of the VAMP2 SNARE motif (NTD; gray arrow), and 3→4 represents irreversible unfolding of the syntaxin SNARE motif (black arrow). 3→4 was accompanied by release of SNAP-25B. Relaxing the resulting Qa-R-SNARE conjugate revealed a featureless FEC, as expected for an unfolded polypeptide (*Figure 2C*, FEC #1, black curve) (*Gao et al., 2012*; *Ma et al., 2015*).

We next asked whether our single-molecule assay could be used to detect and characterize the predicted template complex (*Figure 2A*). The addition of 2 μM Munc18-1 had little effect on the unfolding pathway of the initial syntaxin/VAMP2/SNAP-25B complex (*Figure 2C*, compare gray curves in FECs #1 and #2; *Video 1*). However, the presence of Munc18-1 had a striking effect on the FEC of the remaining Qa-R-SNARE conjugate. Specifically, relaxing (*Figure 2C*, #2, black trace) and then pulling (*Figure 2C*, #3, blue trace) the Qa-R-SNARE conjugate revealed two Munc18-1-dependent features (*Figure 2—figure supplement 2*). In about 40% of the FECs, we observed a small flickering signal at 10–15 pN (*Figure 2C*, #2, blue rectangle; *Figure 2—figure supplement 3*). We attribute this transition (5↔6) to the reversible folding/unfolding of the partially closed syntaxin conformation induced by Munc18-1 (state 6 in *Figure 2D*). More importantly, in about 50% of the FECs, we observed prominent flickering signals at 3–7 pN (*Figure 2C*, #2–3, blue ovals). In a given FEC, these transitions could be sequential (#2, 5↔6 followed by 6↔7) or cooperative (#3, 5↔7). As described in detail below, extensive evidence supports the conclusion that this transition (6↔7 or

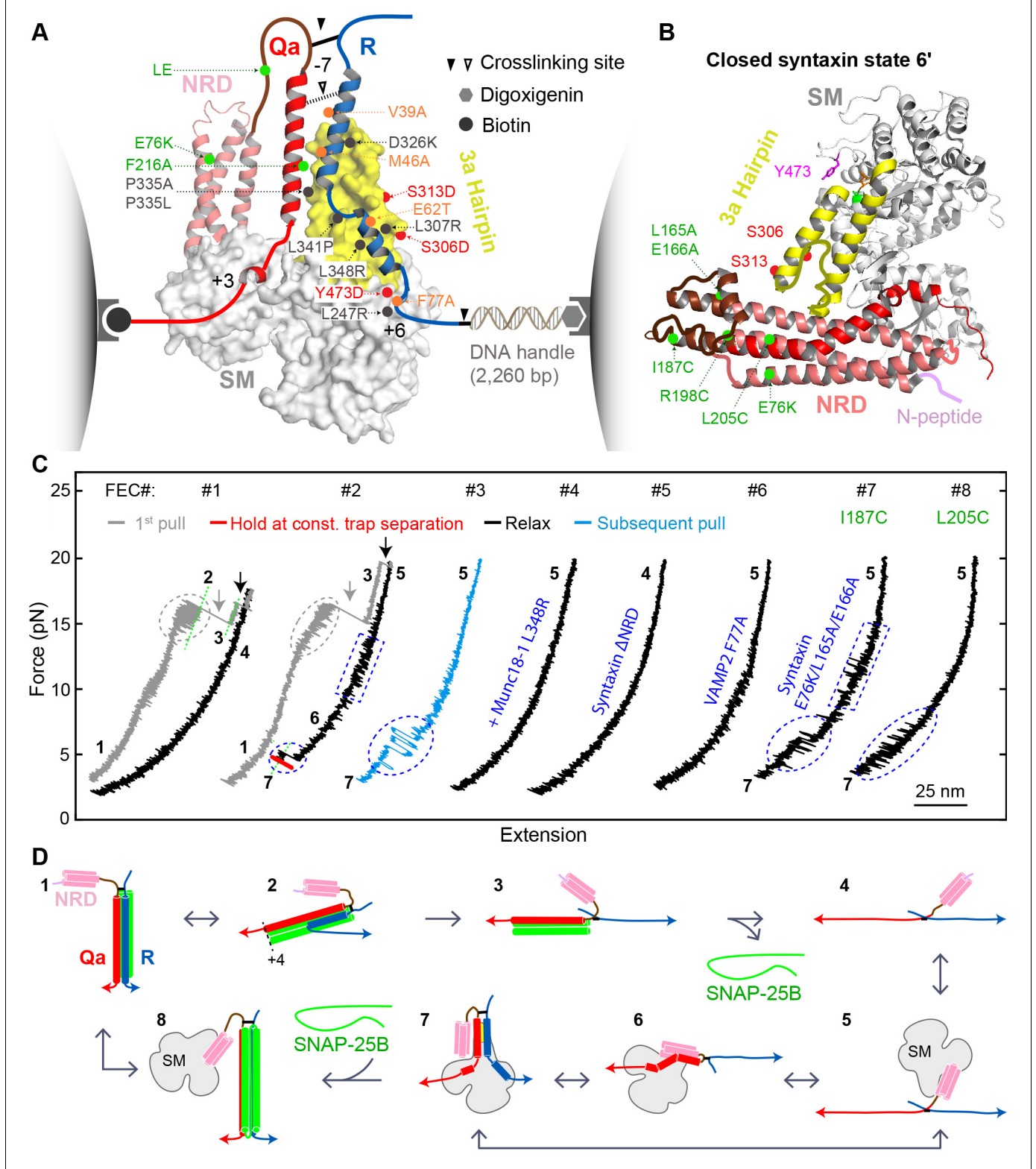

**Figure 2.** Single-molecule manipulation based on optical tweezers revealed a ternary template complex. (**A**) Experimental setup and structural model of the template complex. Some key mutations tested in this study are indicated by dots: red (phosphomimetic mutations) or gray (others) for Munc18-1, green for syntaxin, and orange for VAMP2. The helical hairpin of Munc18-1 domain 3a is highlighted in yellow. The NRD of syntaxin comprises an N-peptide (a.a. 1–26, see B), a three-helical H$_{abc}$ domain (27–146, deep salmon), and a linker region (147–199, brown). The structural model of the template complex is derived from a similar model of Vps33:Vam3:Nyv1 (**Baker et al., 2015**) by extending the N-terminal helix of the R-SNARE to

*Figure 2 continued on next page*

eLIFE Research article                                            Cell Biology | Structural Biology and Molecular Biophysics

*Figure 2 continued*

the −7 layer, as justified herein. The NRD stabilizes the template complex, but its positioning in this model is arbitrary. See also *Figure 2—figure supplement 1*. (B) Crystal structure of closed syntaxin bound to Munc18-1 (PDB ID 3C98) (*Misura et al., 2000*). Highlighted are crosslinking sites (I187, R198, and L205), sites of mutations used to destabilize closed syntaxin (E76K, L165A, and E166A, green dots), and sites of phosphomimetic mutations (red dots and Y473). (C) Force-extension curves (FECs) obtained in the absence (#1) or presence (other FECs) of Munc18-1 in solution. Throughout the figures, all FECs are color coded in the same fashion: gray for pulling the initial purified SNARE complex, blue for subsequent pulls, black for relaxation, and red for holding the system at constant force. The states associated with different extensions (marked by green dashed lines as needed) are numbered as in *Figure 2D*. CTD transitions are indicated by gray ovals, NTD unfolding by gray arrows, t-SNARE unfolding by black arrows, syntaxin transitions by blue rectangles, and template complex transitions by blue ovals. See also *Figure 2—figure supplement 2* and *Figure 2—source datas 1* and *2* in Dataset 1. (D) Schematic diagrams of different states: 1, fully assembled SNARE complex; 2, half-zippered SNARE bundle; 3, t-SNARE complex; 4, fully unfolded SNARE motifs; 5, unfolded SNARE motifs with Munc18-1 bound; 6, partially closed syntaxin; 7, template complex; and 8, Munc18-1-bound assembled SNARE complex. In states 2 and 3, the t-SNARE complex is ordered up to the +4 layer (*Gao et al., 2012*; *Ma et al., 2015*; *Zhang et al., 2016b*). The states are numbered according to the same convention throughout the text and figures.

DOI: https://doi.org/10.7554/eLife.41771.003

The following source data and figure supplements are available for figure 2:

**Source data 1.** MATLAB figure for the FECs shown in *Figure 2C*.

DOI: https://doi.org/10.7554/eLife.41771.011

**Source data 2.** Complete time-dependent instantaneous force, extension, and trap separation obtained in a representative single-molecule experiment in the presence of WT 2 µM Munc18-1.

DOI: https://doi.org/10.7554/eLife.41771.012

**Figure supplement 1.** Sequences, domains, and crosslinking sites of the SNARE proteins used in this study.

DOI: https://doi.org/10.7554/eLife.41771.004

**Figure supplement 2.** Time-dependent extension (top panel), force (middle panel), and trap separation (bottom panel) for a typical experiment to test template complex formation (*Figure 2—source data 2*).

DOI: https://doi.org/10.7554/eLife.41771.005

**Figure supplement 3.** Extension-time trajectories at two constant mean forces showing the opening-closing transition of the partially closed syntaxin molecule.

DOI: https://doi.org/10.7554/eLife.41771.006

**Figure supplement 4.** Force-dependent syntaxin opening probabilities (top panel) and opening and closing rates (bottom panel) obtained by pulling syntaxin from the two N-terminal sites, R198C and I187C (*Figure 2—figure supplement 1*).

DOI: https://doi.org/10.7554/eLife.41771.007

**Figure supplement 5.** FECs of Qa only (#1) or the Qa-R SNARE conjugate (other FECs) pulled from Site I187C in the presence of 2 µM Munc18-1 without (-) or with (+) 60 nM SNAP-25B.

DOI: https://doi.org/10.7554/eLife.41771.008

**Figure supplement 6.** Extension-time trajectories at two constant mean forces showing the opening-closing transition of the syntaxin molecule pulled from the crosslinking site I187C (*Figure 2—figure supplement 1*).

DOI: https://doi.org/10.7554/eLife.41771.009

**Figure supplement 7.** Extension-time trajectories showing conformational transitions of the template complex transition pulled from Site I187C in the absence (top trace) and presence (bottom) of 60 nM SNAP-25B.

DOI: https://doi.org/10.7554/eLife.41771.010

---

5↔7) results from the reversible, cooperative formation and unfolding of the predicted template complex (*Figure 2A*; state 7 in *Figure 2D*). For example, the probability of observing this transition was greatly reduced when either the Munc18-1:VAMP2 interaction or the Munc18-1:syntaxin interaction was abrogated (*Burkhardt et al., 2008*; *Parisotto et al., 2014*) (*Figure 2C*, #4–5; *Table 1*).

## Stability and conformation of the template complex

To examine the stability and folding/unfolding kinetics of the template complex, we monitored the 6↔7 transition over a range of constant mean forces (*Figure 3A–C*; *Video 1*). Detailed analyses of the extension trajectories (*Figure 3A*) revealed the force-dependent unfolding probability and transition rate of the template complex (*Figure 3C*). Specifically, the midpoint of the template complex folding/unfolding transition occurs at 5.1 ± 0.1 pN (mean ± SEM throughout the text) with an associated extension change of 5.42 ± 0.08 nm. Extrapolating the force-dependent measurements to zero force using a nonlinear model (*Gao et al., 2012*; *Rebane et al., 2016*; *Zhang et al., 2016a*) (see also Materials and methods), we obtained the unfolding energy (5.2 ± 0.1 $k_BT$ or 3.1 ± 0.1 kcal/mol) and lifetime (1.4 s) of the template complex (*Figure 3B*). Comparable analysis of the folding/

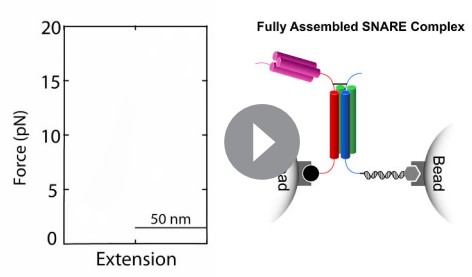

**Video 1.** SNARE complex unfolding and subsequent template complex formation as inferred from single-molecule measurements. The proposed state transitions associated with FEC #2 in *Figure 2C* or *Figure 2—figure supplement 2* are simulated.
DOI: https://doi.org/10.7554/eLife.41771.013

unfolding of the partially closed syntaxin (5↔6) allowed us to estimate its unfolding energy as well (2.6 ± 0.2 $k_B$T; *Figure 2—figure supplement 4*). The extension changes associated with these transitions (5→6→7) are consistent with a structural model of the template complex based on the crystal structures of Vps33:Nyv1 and Vps33:Vam3 (*Baker et al., 2015*) coupled with a worm-like chain model for the unfolded regions of the SNARE proteins and the DNA handle (*Marko and Siggia, 1995*; *Rebane et al., 2016*) (*Figure 2A*). Importantly, the same template complex was observed when we used an alternative Qa-R-SNARE crosslinking site at syntaxin I187C and VAMP2 N29C, but only in conjunction with additional NRD mutations E76K, L165A, and E166A to destabilize the closed conformation of syntaxin (*Figure 2C*, #7; *Figure 2B*; *Figure 2—figure supplements 5–7*). Thus, observation of the template complex was independent of the crosslinking site, requiring only that the closed conformation be destabilized (*Hu et al., 2011*) (see further analysis in the section 'Estimation of the affinity between VAMP2 and Munc18-bound syntaxin in the absence of crosslinking' in Materials and methods).

We used a battery of mutant proteins to test our structural model of the template complex in greater detail (*Figure 2A*). A salient feature of the model is the pivotal role played by a pair of α-helices (a.a. 298–359) within domain 3a of Munc18-1 (*Baker et al., 2015*; *Sitarska et al., 2017*) (yellow in *Figure 2A,B*). These α-helices form an extended helical hairpin that interacts extensively with the NTD of syntaxin and with both the NTD and the CTD of VAMP2 (*Figure 2A* and *Figure 2—figure supplement 1*). Many domain 3a mutations within (L307R, P335L, L341P, L348R) or adjacent (L247R, T248G) to the helical hairpin destabilized the template complex (*Figures 3D* and *4A*, and *Figure 3—figure supplement 1*; *Table 1* and references therein). An internal deletion that removes the distal portion of the helical hairpin (Munc18-1 Δ324–339) abolished formation of the template complex altogether. Notably, two helical hairpin mutations – D326K and P335A – actually stabilized the template complex; both of these mutations are associated with enhanced Munc18-1 function in vitro and in vivo (*Munch et al., 2016*; *Parisotto et al., 2014*; *Sitarska et al., 2017*). Three phosphomimetic mutations (S306D, S313D, and Y347D) are discussed later. As judged by circular dichroism spectropolarimetry, none of the mutations we tested had a significant effect on the overall structure of Munc18-1 (*Figure 3—figure supplement 2*). Overall, the consequences of Munc18-1 mutations are consistent with our structural model.

Reciprocally, we investigated the impact of SNARE motif mutations that appeared likely to affect the SNARE:Munc18-1 interface. Although VAMP2 M46A did not have a significant effect, the rest (VAMP2 S61D/E62T, E62T, Q76A, and F77A; syntaxin F216A, I230G/D231/R232G, and I233G/E234G/Y235G) all destabilized the template complex (*Figure 3E and 4A*; *Table 1*; *Figure 3—figure supplement 1*). The VAMP2 residue Phe 77, located at the so-called +6 layer (*Figure 2—figure supplement 1*), appears to play an especially important role. In our model of the template complex, the side chain of Phe 77 inserts into a deep, hydrophobic pocket in domain 3a, with Leu 247 and Thr 248 residues at the bottom (*Figure 3F*). Phe 77 is highly conserved among R-SNAREs, whereas Leu 247 and Thr 248 are highly conserved among SM proteins (*Figure 3G*). Substituting Phe 77 with Ala dramatically reduced the formation probability of the template complex to 0.06 and its unfolding energy to 1.5 ± 0.3 $k_B$T, the lower limit of our assay (*Figure 2C*, #6; *Figure 3E*; *Figure 3—figure supplement 1*). Similarly, Munc18-1 mutations in the hydrophobic pocket strongly impaired (for L247R or T248G) or totally abolished (for L247A and T248G together) formation of the template complex (*Figure 3D*; *Figure 4A*; *Table 1*). Taken together, our mutagenesis results confirm that the stability of the template complex depends on extensive interactions between Munc18-1 and the two SNARE motifs, including a key anchoring role for the +6 layer Phe of VAMP2.

**Table 1.** Properties of the neuronal template complex.

| SNARE or SM | Mutation or truncation | Unfolding energy ($k_B$T) | Equilibrium force* (pN) | Folding rate ($s^{-1}$) | Unfolding rate ($s^{-1}$) | Partially closed syntaxin[†] Prob. | Template formation Prob.[‡] | N[§] | SNAP-25 binding Prob.[¶] | N[**] |
|---|---|---|---|---|---|---|---|---|---|---|
| WT | - | 5.2 (0.1) | 5.1 (0.1) | 132 | 0.7 | 0.4 | 0.5 | 346 | 0.7 | 50 |
| Munc18-1 | L247R | 1.6 (0.3) | 2.3 (0.1) | - | - | 0.3 | 0.3 | 99 | 0.7 | 6 |
| | T248G | 2.9 (0.2) | 3.1 (0.1) | - | - | 0 | 0.3 | 155 | 0.3 | 16 |
| | L247A/ T248G | <1.5*** | - | - | - | 0 | 0 | 241 | - | - |
| | S306D[¶¶] | 5.8 (0.1) | 5.6 (0.1) | 184 | 0.6 | 0.4 | 0.9 | 123 | 0.9 | 53 |
| | L307R | 4.1 (0.2) | 4.6 (0.1) | | | 0.07 | 0.43 | 114 | 0.58 | 19 |
| | S313D[¶¶] | 6.1 (0.2) | 5.7 (0.1) | 568 | 1.5 | 0.4 | 1 | 162 | 0.8 | 70 |
| | Δ324–339[††,‡‡] | <1.5*** | | - | - | 0 | 0 | 105 | 0 | 0 |
| | D326K[¶¶] | 6.5 (0.2) | 5.7 (0.1) | 420 | 0.6 | 0.03 | 0.9 | 103 | 1 | 27 |
| | P335A[§§] | 6.0 (0.3) | 5.9 (0.1) | 258 | 0.5 | 0.02 | 0.7 | 155 | 0.9 | 11 |
| | P335L[§§] | 4.3 (0.1) | 4.8 (0.1) | 17 | 0.2 | 0.4 | 0.3 | 224 | 0.8 | 36 |
| | L341P[§§] | <1.5*** | | - | - | 0.06 | 0.04 | 176 | 0.5 | 4 |
| | L348R[††,‡‡] | <1.5*** | | - | - | 0.02 | 0.04 | 222 | 0.7 | 6 |
| | Y473D[‡‡] | 4.0 (0.1) | 4.3 (0.2) | - | - | 0 | 0.1 | 395 | 0.5 | 24 |
| VAMP2 | L32G/Q33G | 3.4 (0.2) | 3.9 (0.1) | 310 | 10 | 0.4 | 0.6 | 170 | 0.06 | 33 |
| | V39D | 3.8 (0.4) | 3.9 (0.2) | 90 | 2 | 0.3 | 0.1 | 175 | 0.8 | 13 |
| | M46A | 5.2 (0.4) | 5.1 (0.2) | 130 | 0.7 | 0.3 | 0.5 | 52 | 0.8 | 13 |
| | E62T[††] | 4.1 (0.2) | 4.8 (0.2) | 107 | 5 | 0.4 | 0.5 | 104 | 0.4 | 23 |
| | S61D/ E62T[††] | 3.6 (0.2) | 4.1 (0.1) | | | 0.4 | 0.7 | 56 | 0.2 | 12 |
| | Q76A[††] | 4.7 (0.2) | 4.8 (0.1) | 166 | 2 | 0.4 | 0.6 | 62 | 0.3 | 12 |
| | F77A[‡‡] | 1.5 (0.3) | 2.3 | - | - | 0.5 | 0.1 | 121 | 0.5 | 6 |
| | A81G/A82G | 5.0 (0.3) | 4.9 (0.2) | 130 | 0.8 | 0.4 | 0.5 | 149 | 0.4 | 42 |
| | Δ85–94 | 5.1 (0.2) | 5.0 (0.1) | 120 | 0.7 | 0.4 | 0.5 | 87 | 0.7 | 29 |
| Syntaxin-1 | ΔNRD[††,‡‡] | <1.5*** | - | - | - | 0 | 0.08 | 105 | 0.2 | 12 |
| | ΔN-peptide[††,‡‡] | 3.2 (0.2) | 4.6 (0.1) | 42 | 2 | 0.03 | 0.5 | 328 | 0.4 | 46 |
| | ΔH$_{abc}$[‡‡] | <1.5*** | - | - | - | 0 | 0.06 | 140 | 0.5 | 4 |
| | L165A/E166A (LE)[¶¶] | 6.7 (0.2) | 6.1 (0.1) | 406 | 0.5 | 0.07 | 0.7 | 83 | 0.9 | 26 |
| | LE/E76K | 6.4 (0.2) | 6.0 (0.2) | 123 | 0.2 | 0.07 | 0.9 | 81 | 0.7 | 30 |
| | I202G/I203G | 3.0 (0.3) | 3.8 (0.1) | 240 | 12 | 0.4 | 0.5 | 177 | 0.4 | 33 |
| | F216A | 3.7 (0.1) | 5.1 (0.1) | 82 | 2 | 0 | 0.6 | 155 | 0.9 | 32 |
| | I230G/D231G/ R232G[†††] | 3.6 (0.2) | 4.3 (0.1) | - | - | 0 | 0.5 | 111 | 0.4 | 7 |
| | I233G/E234G/ Y235G[†††] | 3.0 (0.2) | 4.1 (0.1) | - | - | 0 | 0.6 | 122 | 0.7 | 30 |
| | V237G/E238G/ H239G | 5.2 (0.2) | 4.9 (0.1) | 124 | 0.7 | 0.01 | 0.3 | 182 | 0.4 | 14 |
| | T251G/K252G | 5.2 (0.1) | 4.9 (0.1) | 126 | 0.7 | 0.5 | 0.8 | 197 | 0.7 | 47 |
| | Δ255–264 | 5.4 (0.2) | 5.1 (0.1) | 140 | 0.6 | 0.5 | 0.5 | 134 | 0.7 | 29 |
| Syntaxin-1 Munc18-1 | L165A/E166A D326K[¶¶] | 6.6 (0.2) | 6.2 (0.1) | 72 | 0.1 | 0.2 | 0.9 | 85 | 0.2 | 11 |

* Mean of two average forces for the unfolded and folded states when the two states are equally populated (**Rebane et al., 2016**). The equilibrium force of the template complex generally correlates with its unfolding energy. The number in parentheses is the standard error of the mean.

† Detected as the syntaxin- and Munc18-1-dependent transition in the force range of 10–15 pN.

‡ Probability per relaxation or pulling measured in the absence of SNAP-25B.

§ Total number of pulling or relaxation FECs acquired, in which transitions of the template complex or syntaxin are scored, including their average equilibrium forces and extension changes.

¶ Probability of SNAP-25B binding and SNARE assembly per relaxation upon formation of the template complex.

** Total number of relaxation FECs containing the template complex transition.

†† Mutation that reduces membrane fusion in vitro (**Parisotto et al., 2014**; **Shen et al., 2010**; **Shen et al., 2007**).

‡‡ Mutation that impairs exocytosis or neurotransmitter release in vivo (**Meijer et al., 2018**; **Munch et al., 2016**; **Walter et al., 2010**).

§§ Mutation associated with epilepsy (**Stamberger et al., 2016**).

¶¶ Mutation that *enhances* membrane fusion in vitro or neurotransmitter release in the cell (**Genc et al., 2014**; **Gerber et al., 2008**; **Lai et al., 2017**; **Munch et al., 2016**; **Parisotto et al., 2014**; **Richmond et al., 2001**).

*** Unfolding energy below the detection limit of our method, estimated to be 1.5 $k_BT$, or not available due to no, infrequent, or heterogeneous template complex transition.

††† In the observed template complex transition, the template complex frequently dwelled in the unfolded state for an unusually long time (**Figure 3—figure supplement 1**). Thus, the transition is no longer two-state.

DOI: https://doi.org/10.7554/eLife.41771.014

In the binary Vps33:SNARE crystal structures we reported previously (**Baker et al., 2015**), only the central regions of each SNARE motif (Qa-SNARE layers −4 to +3; R-SNARE layers −4 to +6) contact the SM template, whereas both ends of each SNARE motif are likely disordered. In the ternary template complex, however, the two SNARE motifs may be correctly zippered all the way to their N-termini. First, −7 layer mutations (VAMP2 L32G/L33G or syntaxin I202G/I203G) destabilized the template complex, as did a −5 layer mutation (VAMP2 V39D) (**Figure 3E,I**; **Figure 4**; **Table 1**). Second, crosslinking the SNAREs at the −6 layer (via syntaxin L205C and VAMP2 Q36C; open arrowhead in **Figure 2A**; **Figure 2—figure supplement 1**) (**Ma et al., 2015**) enhanced the probability of observing the template complex to 0.93 (**Figure 2C**, #8; **Figure 3H**). Taken together, these data suggest that the −6 layer is properly aligned in the template complex and that the N-terminal regions from layers −7 to −5 – which are unlikely to contact Munc18-1 but nevertheless contribute to the stability of the complex – are correctly zippered (**Figure 2A**). By contrast, altering C-terminal regions of the SNARE motifs (VAMP2 A81G/A82G or Δ85–94; syntaxin V237G/E238G/H239G, T251G/K252G, or Δ255–264) did not affect the stability of the template complex (**Figure 4** and **Table 1**). Thus, syntaxin regions C-terminal to the +3 layer, and VAMP2 regions C-terminal to the +6 layer, are likely disordered in the template complex.

## Template complex facilitates SNARE assembly

To investigate a potential role for the template complex in SNARE assembly, we repeatedly relaxed and pulled the neuronal Qa-R-SNARE conjugate in the presence of SNAP-25B and, where indicated, Munc18-1. During relaxation, we held the Qa-R-SNARE conjugate at constant mean forces, typically around the equilibrium force of the template complex (**Table 1**), for up to 60 s to afford an opportunity for SNAP-25B binding and SNARE assembly. The probability of SNARE assembly per relaxation was measured to assess the efficiency of SNARE assembly. Properly assembled SNARE complexes were taken to be those that: (i) displayed the same extension at low force as the initial purified complex, and (ii) exhibited stepwise unfolding in subsequent rounds of pulling. In the presence of 60 nM SNAP-25B but no Munc18-1, the SNAREs rarely assembled, with a probability of only 0.08 per relaxation (**Figures 5A** and **6**). Increasing the SNAP-25B concentration to 200 nM increased the assembly probability to 0.41 (**Figure 5B**, #2–4; **Figure 6**). This 'spontaneous' (i.e. Munc18-1-independent) SNARE assembly is observed as a single extension drop in the force range of 2–4 pN (**Figure 5C**). The t-SNARE complex forms in the same force range (**Zhang et al., 2016b**), indicating that the spontaneous SNARE assembly we observe here is likely limited by t-SNARE formation. Consistent with the interpretation that the t-SNARE complex is an obligate intermediate in spontaneous SNARE assembly, zippering between a pre-formed t-SNARE complex and the v-SNARE (t-v zippering) occurs at much higher force (**Figure 5B**, #1). Notably, SNAREs misassembled in the absence of

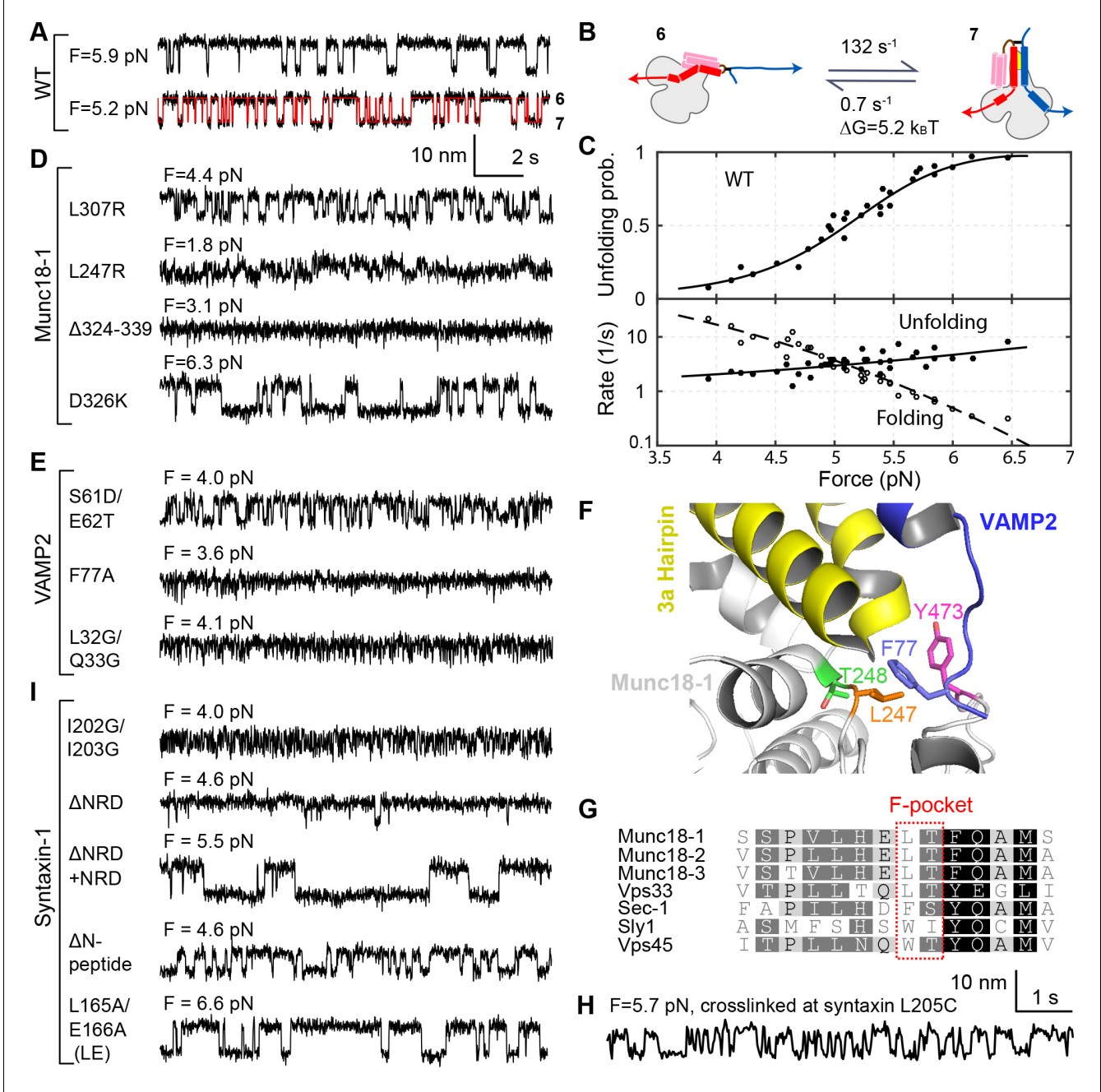

**Figure 3.** Stability, conformation, and folding kinetics of the template complex. (A, D, E, I) Extension-time trajectories at constant mean forces with the WT template complex (A) or its variants containing indicated mutations in Munc18-1 (D), VAMP2 (E), or syntaxin (I). The red trace in A shows an exemplary idealized trajectory derived from hidden Markov modeling. Trajectories in A, D, E, and I share the same scale bars. See also *Figure 3— figure supplement 1* and *Figure 3—source datas 1* and *2* in Dataset 1. (B) Diagram illustrating the transition between the partially closed syntaxin state (state 6 in *Figure 2D*) and the template complex state (state 7); rates and energies are derived from panel C. (C) Force-dependent unfolding probabilities (top) and transition rates (bottom). Best model fits (solid and dashed curves) reveal the stability and folding and unfolding rates of the template complex at zero force (*Figure 4*, *Table 1*, and *Figure 3—source data 3* in Dataset 1). (F) Structural model of VAMP2 F77 anchored in the F-pocket in Munc18-1 composed of L247 and T248, which is covered by Y473. The model was derived by superimposing the structures of Munc18-1: syntaxin (*Figure 2B*; 3C98) and Vps33:Nyv1 (5BV0). (G) Sequence alignment showing F-pocket sequence conservation among SM proteins. (H) Extension-time trajectory of the WT template complex at 5.7 pN. The Qa-R SNAREs were crosslinked between syntaxin L205C and VAMP2 Q36C (*Figure 2A*, open arrowhead). See also *Figure 2—figure supplement 1* and *Figure 3—source data 2* in Dataset 1.

DOI: https://doi.org/10.7554/eLife.41771.015

The following source data and figure supplements are available for figure 3:

*Figure 3 continued on next page*

*Figure 3 continued*

**Source data 1.** MATLAB figure corresponding to *Figure 3A* with an additional trace at force F = 5.0 pN.
DOI: https://doi.org/10.7554/eLife.41771.019
**Source data 2.** MATLAB figure containing expanded traces shown in *Figure 3D,E,I,H*.
DOI: https://doi.org/10.7554/eLife.41771.020
**Source data 3.** MATLAB figure for *Figure 3C*.
DOI: https://doi.org/10.7554/eLife.41771.021
**Figure supplement 1.** FECs obtained in the presence of 2 μM Munc18-1.
DOI: https://doi.org/10.7554/eLife.41771.016
**Figure supplement 2.** Circular Dichroism (CD) spectra show that the mutations tested in our experiments do not significantly alter Munc18-1 folding.
DOI: https://doi.org/10.7554/eLife.41771.017
**Figure supplement 3.** Snapshots of the extension-time trajectories at constant mean forces showing sporadic folding of the template complex in the absence of syntaxin NRD.
DOI: https://doi.org/10.7554/eLife.41771.018

Munc18-1 with a probability of ~0.1, as judged by premature unfolding of the incorrectly assembled complex at low force upon subsequent pulling (*Figure 5B*, #5; *Figure 6*).

The addition of 2 μM Munc18-1 significantly promoted SNARE assembly in the presence of 60 nM SNAP-25B. In a representative experiment, a single Qa-R SNARE conjugate assembled into a proper SNARE complex during each of five consecutive rounds of pulling and relaxation (*Figure 7A*, #1-#5). Overall, based on 67 pulling and relaxation FECs conducted using 15 Qa-R SNARE conjugates, proper assembly was observed with a probability of 0.53 per relaxation (*Figure 6*). However, SNARE assembly tended to occur consecutively: the conditional probability of observing one SNARE assembly event after another such event was 0.79 (N = 52) (*Figure 7A*, #1-#5), likely mediated by a single Munc18-1 molecule. Every SNARE assembly event was accompanied by a SNAP-25B-dependent, 5.5 ± 0.3 nm (N = 50) extension drop from an intermediate state (*Figure 7B,a–c*; *Video 2*). This intermediate had the same average extension relative to the unfolded state 5, the same equilibrium force, and the same response to mutations as the template complex (*Figure 7A*, #6–7; *Figure 7B,d*; *Figure 7—figure supplements 1* and *2*). We conclude that in the presence of Munc18-1, the pre-assembled template complex is required for SNAP-25B binding and SNARE assembly. This conclusion remains valid when the SNAP-25B concentration was increased to 200 nM (*Figure 7B,e*). In this case, the probability of Munc18-1-chaperoned SNARE assembly increased to 0.68 per relaxation (N = 38) (*Figure 6*). In addition, at this higher SNAP-25B concentration, we observed two instances of spontaneous SNARE assembly – that is, assembly not preceded by the formation of the template complex. Importantly, however, the probability of spontaneous assembly events in the presence of Munc18-1 (0.036) was more than ten-fold lower than the probability of spontaneous assembly events in the absence of Munc18-1 (0.41; see above). Thus, Munc18-1 both promotes SNARE assembly via the template complex and inhibits SNARE assembly via the t-SNARE complex (*Figure 6*).

The template complex greatly accelerated proper SNARE assembly. SNAP-25B bound to the template complex with probabilities of 0.71 and 0.84 per relaxation at 60 nM and 200 nM SNAP-25B, respectively, yielding a binding rate constant of ~5 × 10$^5$ M$^{-1}$s$^{-1}$. The rate constant is 25-fold greater than that observed in the absence of Munc18-1 (~2 × 10$^4$ M$^{-1}$s$^{-1}$), presumably because Munc18-1 pre-aligns the N-terminal portions of the syntaxin and VAMP2 SNARE motifs for recognition by SNAP-25B. Consistent with this view, the VAMP2 −7 layer mutations L32G/L33G nearly abolished SNAP-25B binding (*Figure 7B,f*; *Table 1*). Notably, we did not observe any misassembly events in the presence of Munc18-1 (*Figure 6*). Thus, Munc18-1 enhanced the speed, and probably the accuracy, of SNARE assembly.

In most cases, the disassembly FECs of the Munc18-1-reassembled SNARE complexes were indistinguishable from those of the initial SNARE complexes (*Figure 7A*, compare blue curves in #3–5 to gray curve in #1), consistent with proper SNARE reassembly. 14% of the time, however, the subsequent pulling FEC revealed unfolding of the VAMP2 CTD at unusually low force of 4–14 pN (*Figure 7A*, #2). These are not misassembly events, because the SNARE complexes show stepwise NTD unfolding and t-SNARE unfolding identical to the properly assembled SNARE complexes. Further work is required to determine the conformation of these complexes.

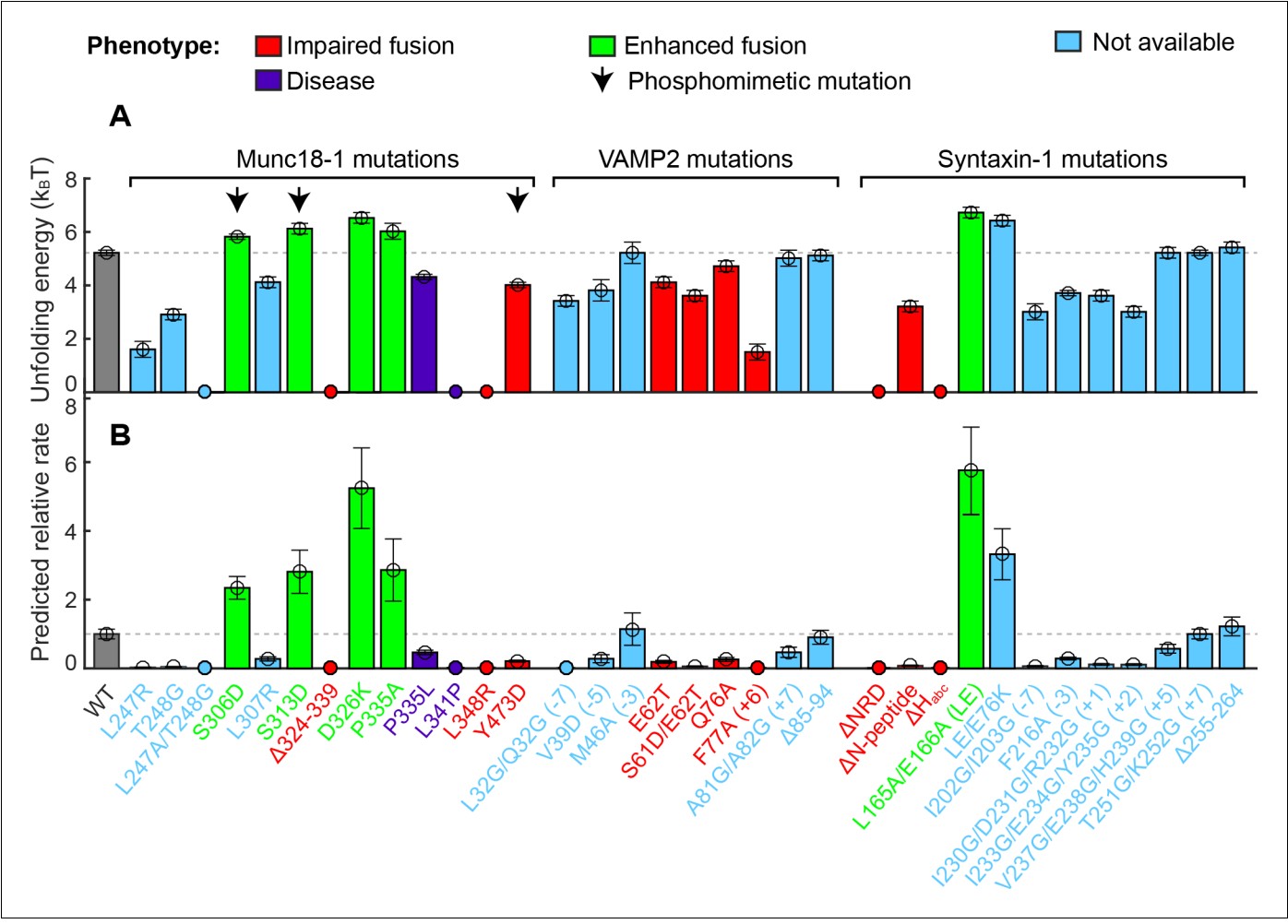

**Figure 4.** Stability of the template complex correlates with SNARE-mediated membrane fusion and neurotransmitter release. (A) Unfolding energy of the WT and mutant template complexes; see also *Table 1*. Unfolding energy that is less than our detection limit (1.5 $k_BT$) is plotted as zero. The unfolding energy is derived from the work required to reversibly unfold the template complex (*Rebane et al., 2016*). The work is measured as the equilibrium force multiplied by the extension change associated with the template complex transition; see also *Figure 3C* and *Figure 3—figure supplement 1*. Numbers in parentheses after SNARE mutant names indicate the layer numbers associated with the corresponding mutations. Error bars indicate standard errors of the mean. See also *Figure 4—source data 1*. (B) Relative rate of SNARE assembly and membrane fusion calculated using *Equation 1*, derived under the assumption that the rate is determined by the stability of the template complex.

DOI: https://doi.org/10.7554/eLife.41771.022

The following source data is available for figure 4:

**Source data 1.** Data summary table for the results shown in *Figure 4*.

DOI: https://doi.org/10.7554/eLife.41771.023

## N-terminal regulatory domain of syntaxin stabilizes template complex

Once initiated, the reversible template complex transition (6↔7 or 5↔7) typically persisted for over 10 min at constant mean force, even after the free Munc18-1 in the solution was removed. We suspected that the persistent association between Munc18-1 and the Qa-R-SNARE conjugate was attributable to the NRD of syntaxin, as suggested by previous results (*Burkhardt et al., 2008*; *Shen et al., 2010*; *Shen et al., 2007*; *Zhou et al., 2013*). Indeed, NRD truncation (ΔNRD) reduced the probability of observing template complex formation from 0.5 to 0.08 (*Figure 2C*, #5), consistent with the idea that the NRD recruits Munc18-1. The average lifetime of the template complex formed by ΔNRD was also shorter (*Figure 3I* and *Figure 3—figure supplement 3*), indicating that the NRD stabilizes the template complex. Unexpectedly, addition of 2 µM NRD in trans was able to rescue the defect caused by ΔNRD: the template complex now formed efficiently (probability = 0.6,

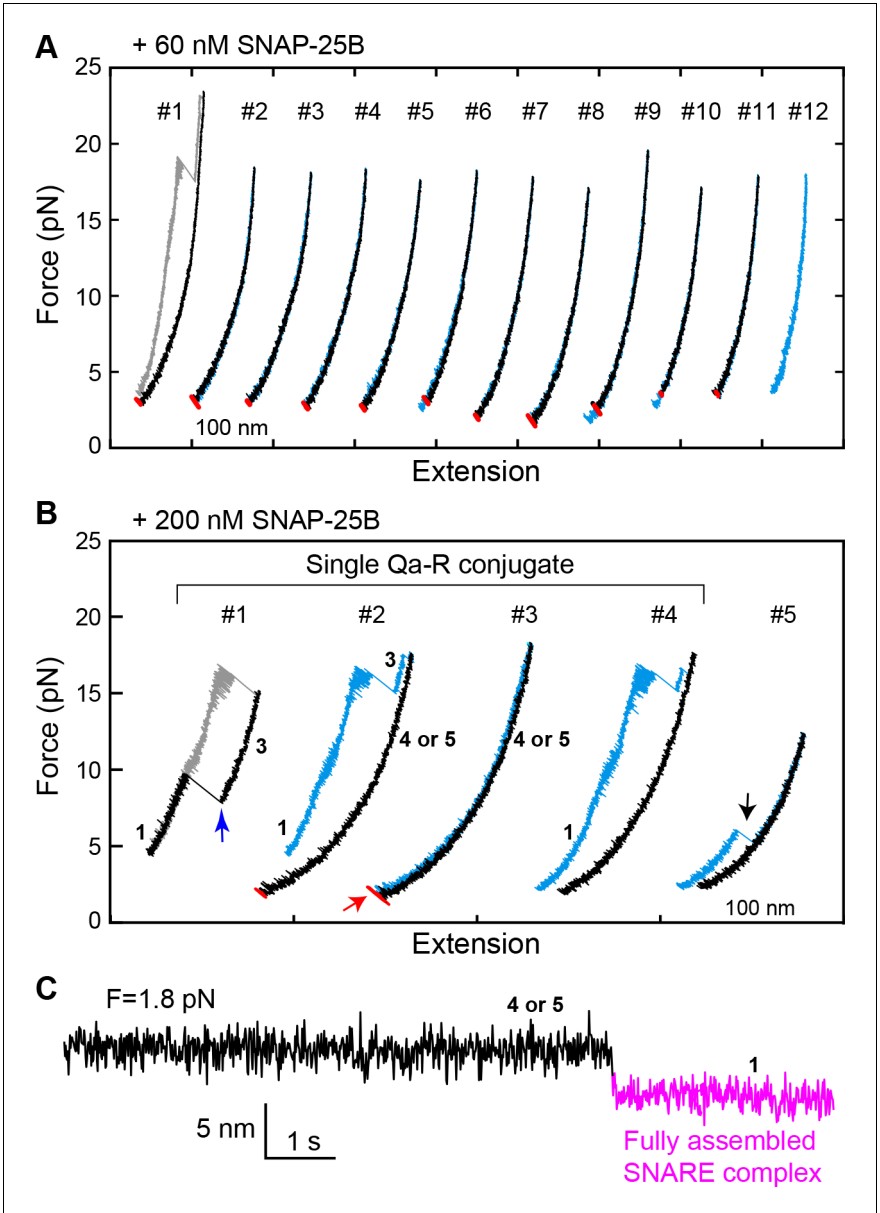

**Figure 5.** Spontaneous SNARE assembly in the absence of Munc18-1 is inefficient. (**A**) Representative FECs obtained by consecutively pulling and relaxing a single Qa-R SNARE conjugate in the presence of 60 nM SNAP-25B (*Figure 5—source data 1*). No SNARE assembly is observed. (**B**) Representative FECs obtained in the presence of 200 nM SNAP-25B showing spontaneous SNARE assembly (#1–4, *Figure 5—source data 2*) and misassembly (#5). FECs #1–4 are from a single Qa-R SNARE conjugate. In FEC #1, the SNARE complex was relaxed just after the t- and v-SNAREs were unzipped to observe t-v zippering at low force (blue arrow). The black arrow marks disassembly of the misfolded SNARE complex. Throughout figures, red arrows indicate SNARE reassembly. (**C**) Extension-time trajectory exhibiting cooperative de novo SNARE assembly at a constant mean force (*Figure 5—source data 3*). Shown here is the spontaneous SNARE assembly observed in panel B marked by the red arrow. Throughout figures, the fully assembled SNARE state (state 1 or 8) is shown in magenta.
DOI: https://doi.org/10.7554/eLife.41771.024

The following source data is available for figure 5:

**Source data 1.** MATLAB figure corresponding to *Figure 5A*.
DOI: https://doi.org/10.7554/eLife.41771.025
**Source data 2.** MATLAB figure corresponding to *Figure 5B* (FEC#1–4).
DOI: https://doi.org/10.7554/eLife.41771.026
**Source data 3.** MATLAB figure corresponding to *Figure 5C*.
DOI: https://doi.org/10.7554/eLife.41771.027

N = 35) at an equilibrium force close to that of the WT template complex, albeit with slower transition kinetics (*Figure 3I* and *Figure 3—figure supplement 1*). Thus, the NRD can bind to and stabilize the template complex in trans.

Next, we dissected the roles of different NRD regions. Removing the 'N-peptide' at the extreme N-terminus of the NRD (*Figure 2B*) destabilized the template complex (*Figure 3I and 4*; *Table 1*), while removing the three-helix bundle $H_{abc}$ domain ($\Delta H_{abc}$) abolished template complex formation altogether (*Table 1* and *Figure 3—figure supplement 1*). By contrast the 'LE' mutation (L165A/E166A in the linker region between the $H_{abc}$ domain and the SNARE motif; see *Figure 2B*) (*Dulubova et al., 1999*), which promoted SNARE assembly as expected (*Burkhardt et al., 2008*; *Gerber et al., 2008*; *Ma et al., 2011*; *Richmond et al., 2001*), stabilized the template complex (*Figures 3I* and *4A*). Taken together, our results imply that the NRD has a three-fold role in template complex formation. When syntaxin is closed (state 6' in *Figure 2B*), the NRD inhibits template complex formation (*Figure 2—figure supplement 5*), as also shown previously (*Burkhardt et al., 2008*). When syntaxin is partially open (state 6 in *Figure 2D*), the NRD recruits Munc18-1 for fast folding of the template complex. Finally, once the template complex has formed (state 7 in *Figure 2D*), the NRD plays a direct stabilizing role. The structural basis for this stabilizing role awaits further investigation.

## Munc18-1 inhibits t-SNARE complex formation

As noted above, Munc18-1 not only promotes SNARE assembly via the template complex, but also inhibits spontaneous SNARE assembly (i.e. assembly not preceded by template complex formation).

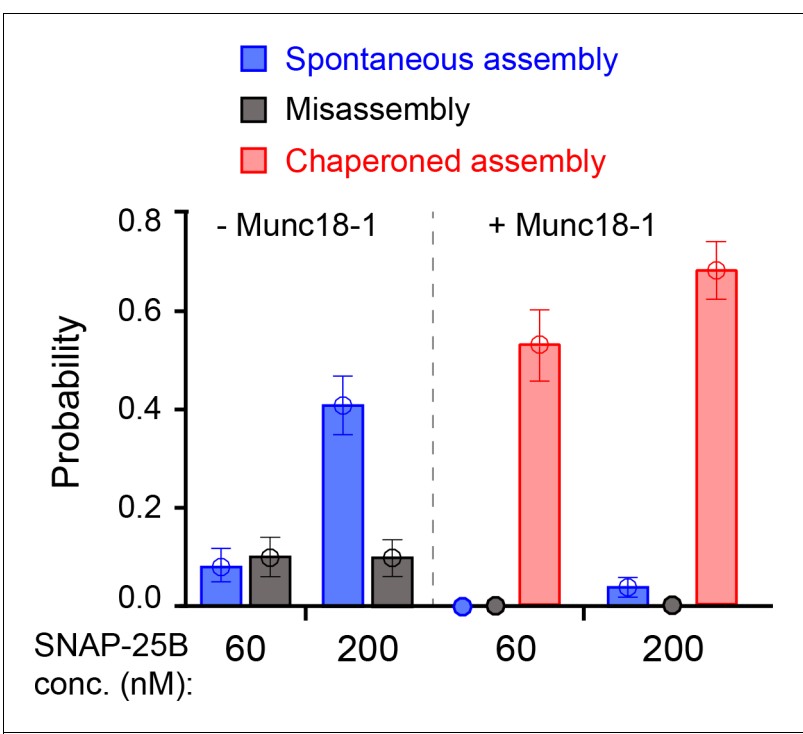

**Figure 6.** Comparison of SNARE assembly in the absence and presence of Munc18-1. Bars indicate probabilities of Munc18-1-independent or spontaneous SNARE assembly (blue), Munc18-1-chaperoned SNARE assembly (red), and SNARE misassembly (black). See also *Figure 6—source data 1*.
DOI: https://doi.org/10.7554/eLife.41771.028

The following source data is available for figure 6:

**Source data 1.** Data summary table for the results shown in *Figure 6*.
DOI: https://doi.org/10.7554/eLife.41771.029

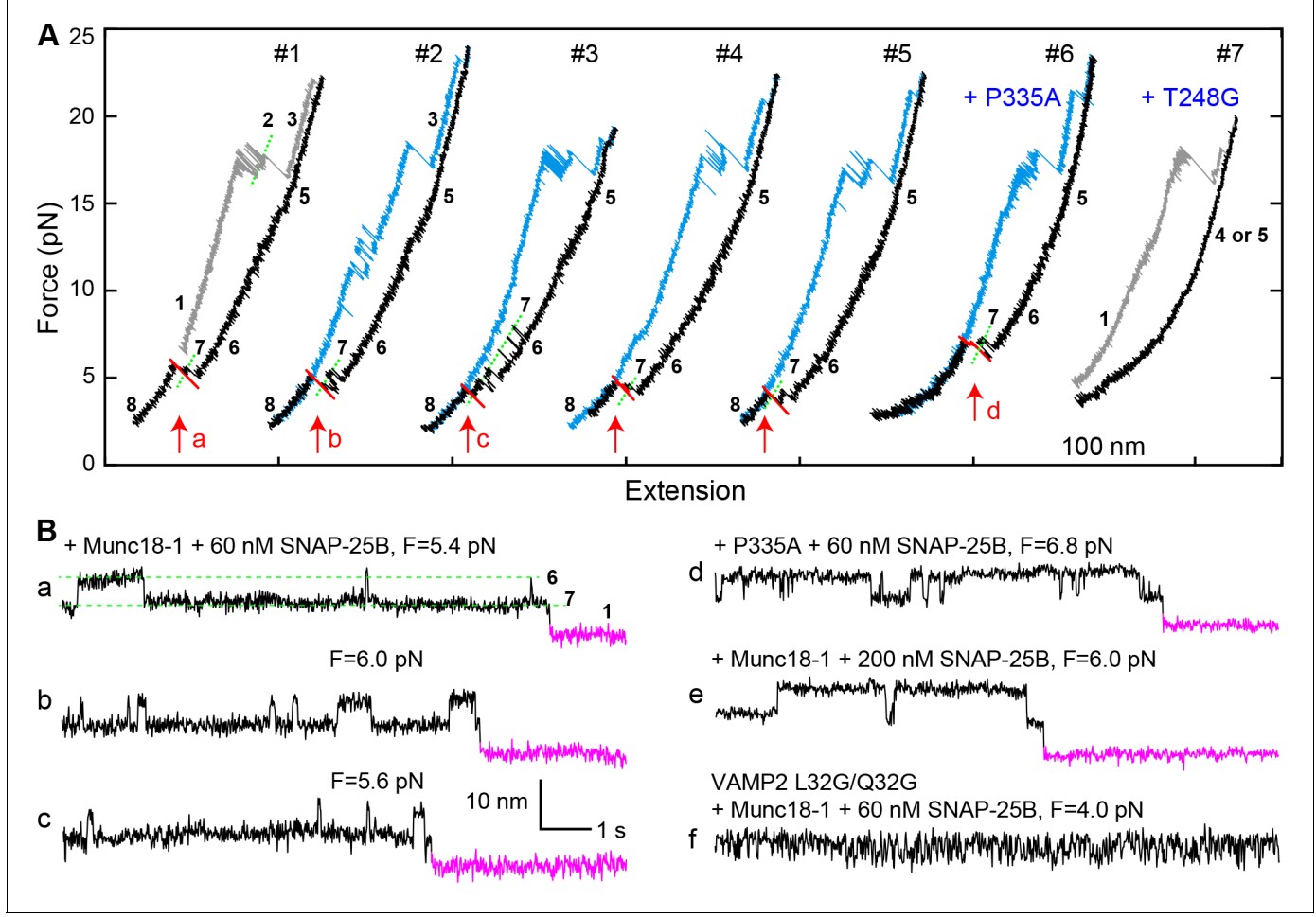

**Figure 7.** Template complex facilitates SNARE assembly. (**A**) Representative FECs obtained in the presence of 60 nM SNAP-25B and 2 µM WT Munc18-1 (#1–5 ) or Munc18-1 mutants P335A (#6) or T248G (#7). FECs #1–5 represent consecutive rounds of manipulation of a single Qa-R SNARE conjugate. See also *Figure 7—figure supplement 1* and *Figure 7—source data 1*. (**B**) Extension-time trajectories at the indicated constant mean forces showing SNARE assembly. Traces a-d were extracted from FEC regions marked with correspondingly labeled red arrows in panel A. Trace f shows rapid template complex transitions without SNAP-25B binding. See also *Figure 7—figure supplement 2* and *Figure 7—source data 2*.
DOI: https://doi.org/10.7554/eLife.41771.030

The following source data and figure supplements are available for figure 7:

**Source data 1.** MATLAB figure corresponding to *Figure 7A* (FECs #1–5).
DOI: https://doi.org/10.7554/eLife.41771.033
**Source data 2.** MATLAB figure corresponding to *Figure 7B*.
DOI: https://doi.org/10.7554/eLife.41771.034
**Figure supplement 1.** FECs obtained in the presence of 2 µM Munc18-1 and 60 nM SNAP-25B in the solution.
DOI: https://doi.org/10.7554/eLife.41771.031
**Figure supplement 2.** Extension-time trajectories at constant mean forces.
DOI: https://doi.org/10.7554/eLife.41771.032

Using a very similar experimental approach, we previously showed that spontaneous SNARE assembly proceeds by a different route (*Gao et al., 2012*; *Zhang et al., 2016b*): first, syntaxin binds SNAP-25B to form a t-SNARE complex; then, VAMP2 zippers with the t-SNARE complex in a process called t-v zippering (*Gao et al., 2012*; *Zhang et al., 2016b*). In the presence of Munc18-1, t-SNARE complexes were never observed. Thus, Munc18-1 appears to inhibit spontaneous SNARE assembly by suppressing formation of the t-SNARE complex intermediate.

Our results argue that Munc18-1 accelerates SNARE assembly by means of an on-pathway template complex intermediate. By contrast, a previous model instead proposed that Munc18-1

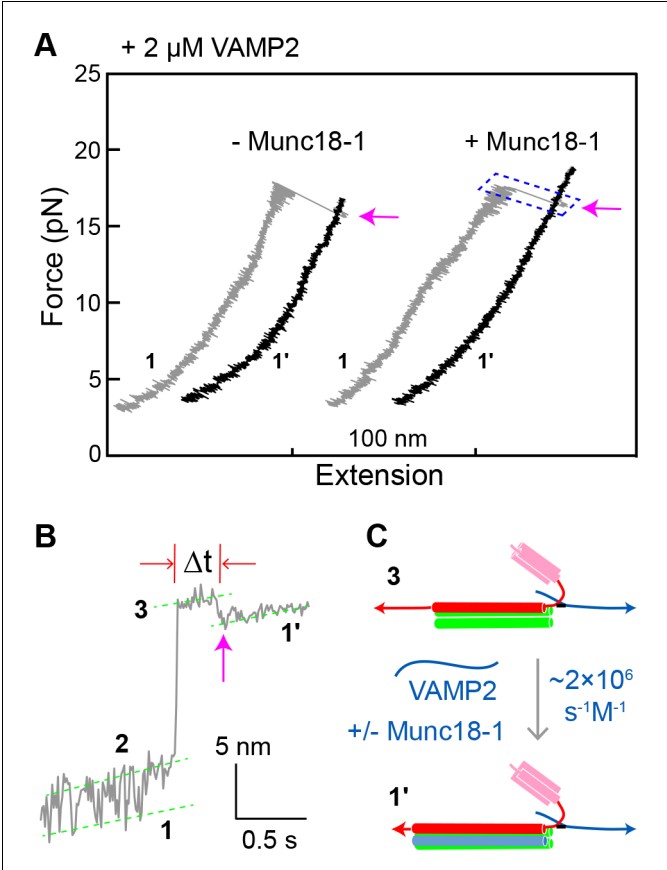

**Figure 8.** Munc18-1 does not significantly accelerate zippering between t- and v-SNAREs. (**A**) FECs obtained by pulling single WT SNARE complexes in 2 µM soluble VAMP2 in the absence or presence of 2 µM Munc18-1. Magenta arrows mark binding of the VAMP2 molecules in the solution to the t-SNARE complexes generated by unzipping the ternary SNARE complex (see panel C). See also *Figure 8—source data 1*. (**B**) Close-up view of an extension-time trajectory displaying VAMP2 binding in trans. The trajectory corresponds to the boxed pulling region in A. As observed previously (*Ma et al., 2015*; *Zhang et al., 2016b*), VAMP2 binding (indicated by the magenta arrow) induced folding of the disordered C-terminus of the t-SNARE complex, decreasing its extension by 2.3 ± 0.1 nm and generating state 1' (see panel C). It took an average time (Δt) of ~0.3 s for the free VAMP2 in the solution to bind the t-SNARE complex. Note that the SNARE complexes in state 1' and state 1 (*Figure 2D*) are pulled from different sites. See also *Figure 8—source data 2*. (**C**) Diagram illustrating VAMP2 induced t-SNARE folding and extension shortening.

DOI: https://doi.org/10.7554/eLife.41771.035

The following source data is available for figure 8:

**Source data 1.** MATLAB figure corresponding to *Figure 8A*.
DOI: https://doi.org/10.7554/eLife.41771.036

**Source data 2.** MATLAB figure corresponding to *Figure 8B*.
DOI: https://doi.org/10.7554/eLife.41771.037

accelerates t-v zippering (*Dawidowski and Cafiso, 2016*; *Jakhanwal et al., 2017*; *Shen et al., 2007*; *Zhang et al., 2016b*). To address this possibility directly, we pulled ternary SNARE complexes to generate the t-SNARE complex (state 3) in the presence of 2 µM soluble VAMP2 (*Figure 8A*). The free VAMP2 molecule rapidly bound the t-SNARE complex (*Figure 8B,C*), with a binding rate constant ($1.6 \times 10^6$ $M^{-1}s^{-1}$) close to a previously published value ($0.5 \times 10^6$ $M^{-1}s^{-1}$) (*Pobbati et al., 2006*). The binding constant was little changed ($2.0 \times 10^6$ $M^{-1}s^{-1}$) by the addition of 2 µM Munc18-1. The average t-v zippering force was ~10 pN with or without Munc18-1 (*Figure 5B*, #1, blue arrow). Thus, we find no evidence to support the model that Munc18-1 promotes t-v zippering.

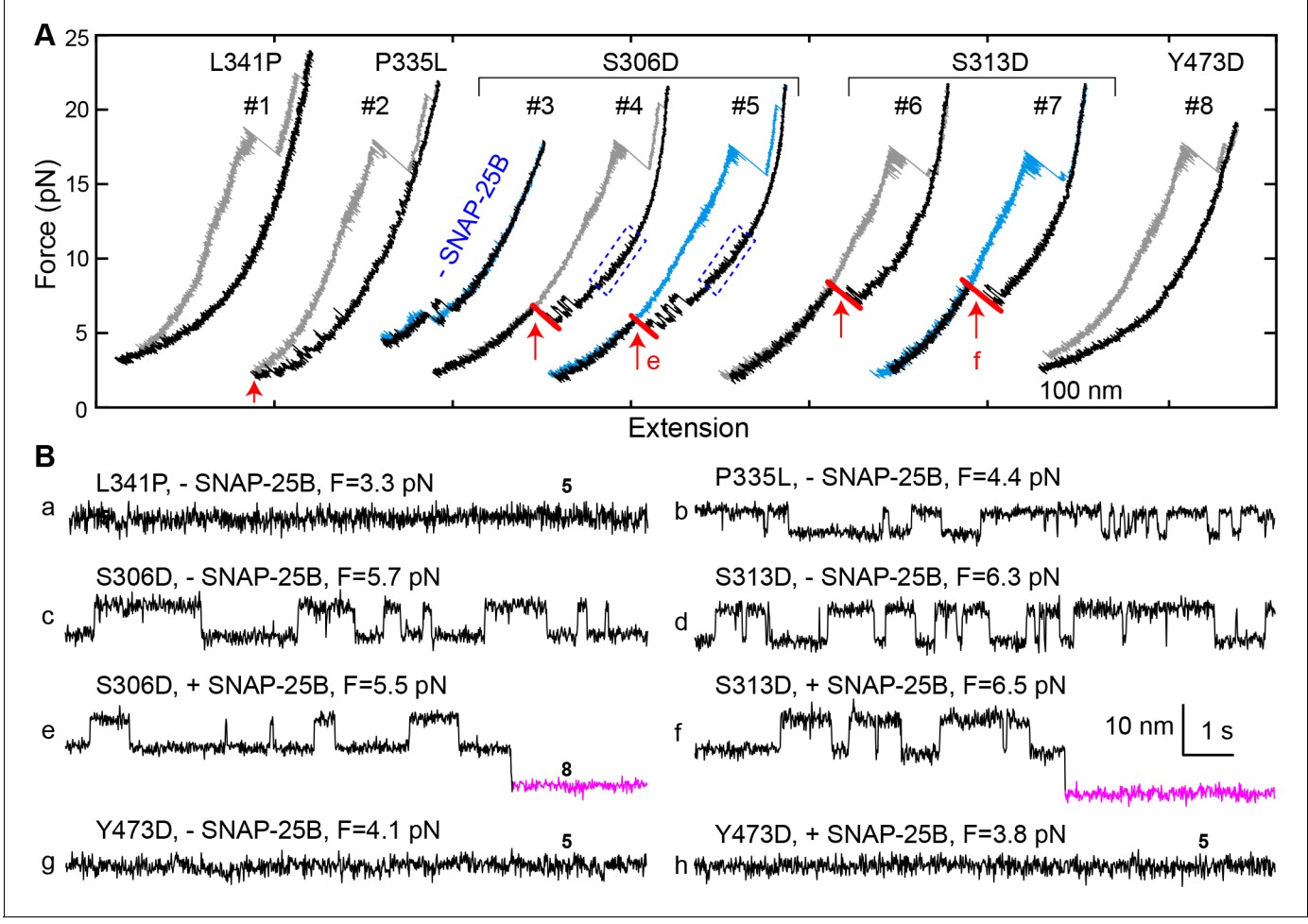

**Figure 9.** Munc18-1 phosphomimetic and disease mutations altered chaperoned SNARE assembly. (**A**) FECs for Munc18-1 mutations with 0 nM (#3) or 60 nM (others) SNAP-25B. See also *Figure 9—source data 1*. (**B**) Extension-time trajectories at the indicated constant mean forces, some of which (**e** and **f**) are extracted from panel A. In panels a, g, and h, no template complex formation is observed. See also *Figure 9—source data 2*.

DOI: https://doi.org/10.7554/eLife.41771.039

The following source data is available for figure 9:

**Source data 1.** MATLAB figure corresponding to *Figure 9A* (FECs #3–7).
DOI: https://doi.org/10.7554/eLife.41771.040
**Source data 2.** MATLAB figure corresponding to *Figure 9B*.
DOI: https://doi.org/10.7554/eLife.41771.041

## Function-altering and phosphomimetic mutations

In examining a large number of SNARE and SM mutations (*Figure 4A* and *Table 1*), we found that modifications known to compromise SNARE assembly, membrane fusion, and/or neurotransmitter release (*Munch et al., 2016*; *Parisotto et al., 2014*; *Shen et al., 2007*; *Walter et al., 2010*; *Zhou et al., 2013*) invariably destabilized the template complex (*Figure 4A*, red). Munc18-1 I341P and P335L are two examples of hundreds of Munc18-1 and SNARE mutations associated with epilepsy and other disorders (*Stamberger et al., 2016*). P335L destabilized, and I341P abrogated altogether, the template complex, leading to impaired Munc18-1-chaperoned SNARE assembly (*Figure 9A*, #1–2; *Figure 9B,a,b*; *Figure 4A*, purple). Conversely, mutations known to increase SNARE assembly, membrane fusion, and/or neurotransmitter release (*Genc et al., 2014*; *Gerber et al., 2008*; *Munch et al., 2016*; *Parisotto et al., 2014*; *Sitarska et al., 2017*) displayed enhanced template complex stability (*Figure 4A*, green bars). This correlation establishes that the template complex is an important intermediate for membrane fusion in vivo.

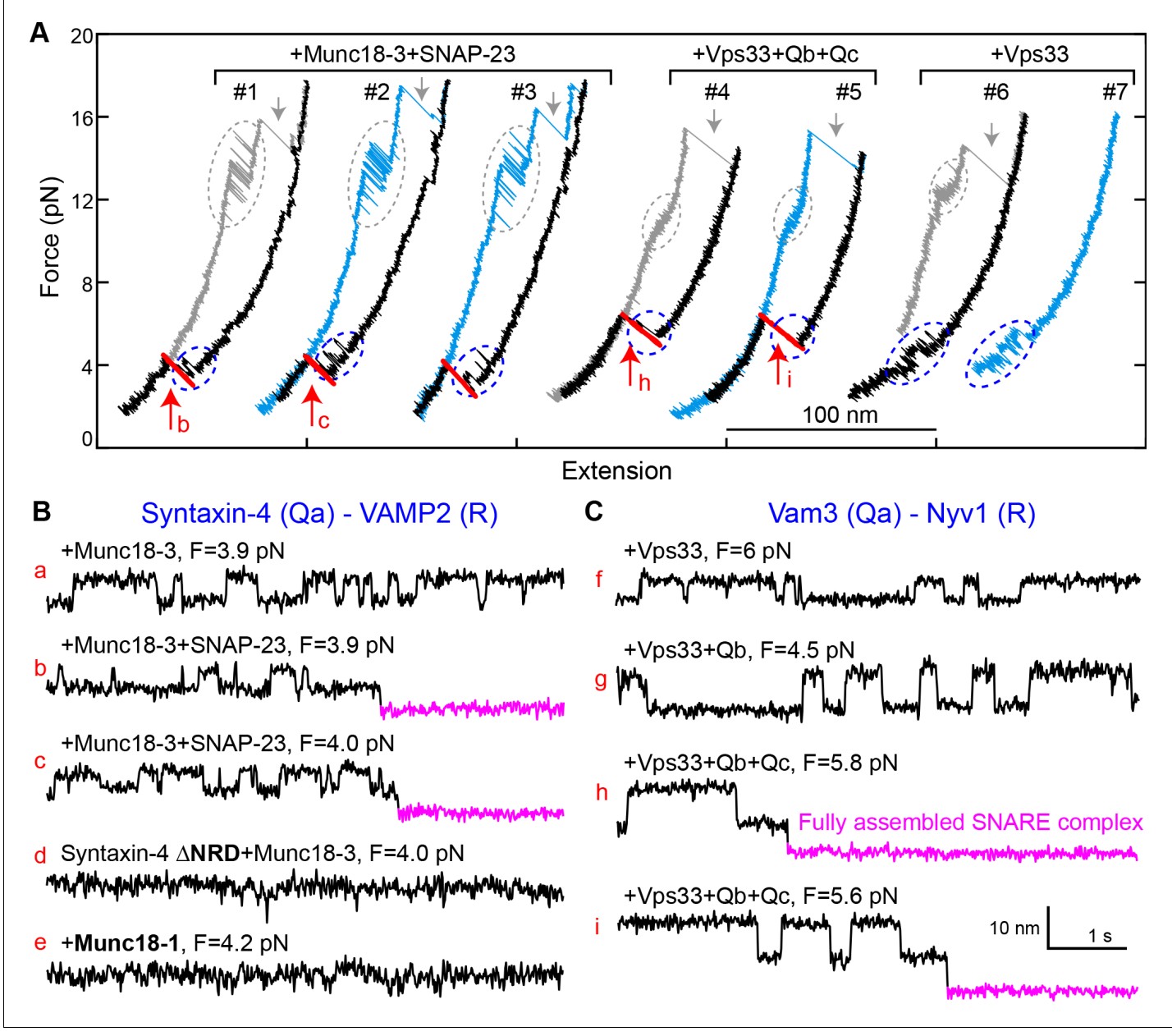

**Figure 10.** Munc18-3 and Vps33 catalyze SNARE assembly via template complexes. (**A**) FECs of the Munc18-3 or Vps33 cognate Qa-R SNARE conjugate in the presence of the indicated protein(s). See also *Figure 10—source datas 1* and *2*. (**B–C**) Extension-time trajectories at the indicated constant mean forces, some of which (b, c, h, and i) are extracted from panel A. See also *Figure 10—source datas 3* and *4*.

DOI: https://doi.org/10.7554/eLife.41771.042

The following source data and figure supplements are available for figure 10:

**Source data 1.** MATLAB figure corresponding to *Figure 10A* (FECs #1–3).
DOI: https://doi.org/10.7554/eLife.41771.046
**Source data 2.** MATLAB figure corresponding to *Figure 10A* (FECs #4–7).
DOI: https://doi.org/10.7554/eLife.41771.047
**Source data 3.** MATLAB figure corresponding to *Figure 10B* (traces a-e).
DOI: https://doi.org/10.7554/eLife.41771.048
**Source data 4.** MATLAB figure corresponding to *Figure 10C* (traces g-i).
DOI: https://doi.org/10.7554/eLife.41771.049
**Figure supplement 1.** FECs obtained by pulling and relaxing a single syntaxin-4-VAMP2 conjugate (#1–4) or Vam3-Nyv1 conjugate (#5–8) in the presence of the indicated protein or proteins.

*Figure 10 continued on next page*

*Figure 10 continued*

DOI: https://doi.org/10.7554/eLife.41771.043

**Figure supplement 2.** FECs displaying Vps33-catalyzed vacuolar SNARE assembly, marked by red arrows.

DOI: https://doi.org/10.7554/eLife.41771.044

**Figure supplement 3.** Probabilities of vacuolar SNARE assembly per relaxation under different conditions.

DOI: https://doi.org/10.7554/eLife.41771.045

The phosphorylation of Munc18 proteins regulates neurotransmitter release and insulin secretion (*de Jong and Verhage, 2009*; *Genc et al., 2014*; *Jewell et al., 2011*; *Meijer et al., 2018*). To explore the mechanism(s) underlying these observations, we examined three phosphomimetic mutations, indicated by arrows in *Figure 4A*. Phosphorylation of domain 3a residues Ser 306 and Ser 313 (*Figure 2A–B*) enhances neurotransmitter release and contributes to short-term memory (*Genc et al., 2014*). Correspondingly, the phosphomimetic mutations S306D and S313D each stabilized the template complex (*Figure 4A*; *Figure 9A*, #3–7) and increased the probabilities of template formation and SNARE assembly (*Table 1*; *Figure 9B,c–f*). Conversely, the phosphomimetic mutation Y473D, which abrogates membrane fusion in vivo (*Meijer et al., 2018*), destabilized the template complex (*Figure 4A*; *Figure 9A*, #8; *Figure 9B,g–h*) and reduced the probabilities of template formation and SNARE assembly (*Table 1*). Tyr 473 is located immediately adjacent to the predicted binding pocket for the +6 layer Phe of VAMP2 (*Figure 3F*) and likely plays a significant role in VAMP2 binding. Taken together, these results suggest that Munc18-1 phosphorylation regulates synaptic vesicle fusion by modulating the stability of the template complex.

Many of the mutations that we tested cause modest changes in the unfolding energy of the template complex (*Figure 4A*) but, seemingly paradoxically, cause large changes in membrane fusion activity (*Genc et al., 2014*; *Munch et al., 2016*; *Parisotto et al., 2014*; *Sitarska et al., 2017*; *Stamberger et al., 2016*). To appreciate the impact of template complex stability on membrane fusion, we estimated the relative rate of membrane fusion mediated by the mutant or WT SNARE-Munc18-1 complex. Our estimation contains three assumptions. First, the template complex is a rate-limiting intermediate for SNARE assembly, and thus the overall rate of SNARE assembly is proportional to the equilibrium probability of the template complex determined by the Boltzmann distribution. Second, the mutation only affects the stability of the template complex, but not SNARE zippering, nor binding of any other regulatory proteins to the SNARE-Munc18-1 complex. Finally, we assume that the overall rate of membrane fusion or exocytosis is proportional to the overall rate of SNARE assembly. With these assumptions, the overall rate of membrane fusion $k$ is proportional to $exp(G) \times r$, where $G$ and $r$ are the unfolding energy of the template complex and the rate of SNAP-25B binding to the template complex, respectively. Consequently, the overall rate of membrane fusion mediated by a mutant SNARE-Munc18-1 complex $k_m$ relative to that of the WT complex $k_{WT}$ can be calculated as

$$\frac{k_m}{k_{WT}} = \frac{r_m}{r_{WT}} exp(G_m - G_{WT}), \qquad (1)$$

where the subscripts $m$ and $WT$ represent mutant and wild-type, respectively. Using this equation, we computed the relative rate of membrane fusion for all of the mutants we tested (*Figure 4B*). Our measurements predict that Munc18-1 mutation P335A increases the overall rate of membrane fusion by 2.9 (±0.9) fold, in good agreement with the observed enhancement of ~2 fold for evoked release in the cell and of 4–5 fold in liposome-liposome fusion in vitro (*Munch et al., 2016*; *Parisotto et al.,*

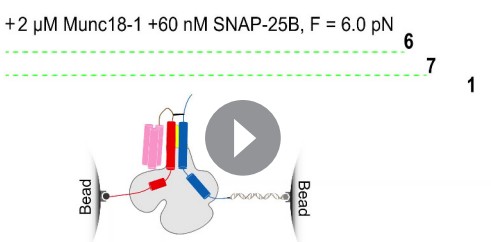

+2 µM Munc18-1 +60 nM SNAP-25B, F = 6.0 pN 6 7 1

**7. Template Complex**

**Video 2.** Template complex facilitates SNAP-25B binding and SNARE assembly. The extension at a constant mean force of 6.0 pN corresponding to trace b in *Figure 7B* and its associated state transition are simulated. For simplicity, only the right bead was simulated to move in response to SNARE conformational changes. In reality, the left bead moved synchronously with the right bead, but in an opposite direction, as shown in *Video 1*.

DOI: https://doi.org/10.7554/eLife.41771.038

*2014*). For Munc18-1 D326K, we predict an increase in the fusion rate by 5 (±1) fold, consistent with the activation of membrane fusion observed in vitro in the absence of calcium triggering: ~4 fold for lipid mixing and ~7 fold for content mixing (*Sitarska et al., 2017*). By contrast, both reconstituted membrane fusion and our prediction show that Munc18-1 L348R abolished membrane fusion (*Parisotto et al., 2014*; *Sitarska et al., 2017*). In conclusion, the concordance between the effect of mutations on membrane fusion or neurotransmitter release and on the stability of the template complex strongly suggests that the template complex is a physiologically relevant intermediate in SNARE assembly.

Finally, we note that VAMP2 and syntaxin mutations, in addition to changing the stability of the template complex, can influence subsequent SNARE zippering. For example, VAMP2 F77A destabilizes both the template complex and the C-terminal region of the SNARE complex (*Figure 3—figure supplement 1*, FEC #25). Layer mutations C-terminal to the +3 layer in syntaxin and the +6 layer in VAMP2 also destabilize the C-terminal region of the SNARE complex, but without affecting the template complex (*Figure 4A*). In general, template complex stability does not predict the impact of given mutation on SNARE zippering. Nonetheless, our measurements make it possible to dissect the respective contributions of Munc18-1 chaperoning and SNARE zippering to the dynamics of membrane fusion.

## Template mechanism is conserved among SM proteins

To generalize our findings, we investigated two other SM proteins, Munc18-3 and Vps33 (*Figure 2—figure supplement 1*). Munc18-3 and its cognate SNAREs syntaxin 4 (Qa), VAMP2 (R), and SNAP-23 (Qbc) mediate fusion of glucose transporter 4- (GLUT4-) containing vesicles with the plasma membrane, promoting glucose uptake (*Bryant and Gould, 2011*; *Yu et al., 2013*). Vps33 and its cognate vacuolar SNAREs Vam3 (Qa), Nyv1 (R), Vti1 (Qb), and Vam7 (Qc) mediate membrane fusion in endolysosomal trafficking (*Wickner, 2010*). Like other SNARE complexes (*Zorman et al., 2014*), the GLUT4 and vacuolar SNARE complexes disassembled stepwise via one or more partially-zippered intermediates (*Figure 10A*). In the absence of SM proteins, spontaneous SNARE assembly was inefficient, with a probability per relaxation of 0.02 for GLUT4 SNAREs (60 nM SNAP-23) and of 0.04 for vacuolar SNAREs (1 µM Vti1 and 1 µM Vam7) (*Figure 10—figure supplement 1*). In the presence of 2 µM Munc18-3 or 0.4 µM Vps33, the probability of SNARE assembly increased to 0.44 for Munc18-3 and to 0.65 for Vps33 (*Figure 10A*, #1–5). Thus, both SM proteins strongly enhance the rate of SNARE assembly. All of the more than 50 Munc18-3-mediated SNARE assembly events we observed in our experiments were mediated by the corresponding template complex (blue ovals in *Figure 10A*, #1–3; *Figure 10B,a–c*). The Munc18-3 template complex displayed a stability (4.3 ± 0.2 $k_BT$) and an extension relative to the unfolded state similar to those of the Munc18-1 template complex. In addition, the Munc18-3 template complex depended on the NRD of syntaxin 4, as NRD truncation reduced the probability of observing the template complex transition to 0.03 (*Figure 10B,d*). Thus, Munc18-3 and Munc18-1 are quantitatively similar in their ability to chaperone cognate SNARE complex assembly. Importantly, neither Munc18-1 nor Munc18-3 could catalyze the assembly of the other's cognate SNAREs (*Figure 10B,e*).

We also observed a Vps33-mediated template complex in the absence of Qb- and Qc-SNAREs (blue ovals in *Figure 10A*, #6–7; *Figure 10C,f*). Relaxing the Qa-R-SNARE conjugate in the presence of both Vps33 and the Qb-SNARE increased the extension change associated with the template complex transition from 4–6 nm to 7–9 nm (*Figure 10C*, compare trace g to trace f), indicating that the Qb-SNARE induced further folding of the templated SNAREs. Relaxing the Qa-R-SNARE conjugate in the presence of both the Qb- and Qc-SNAREs triggered assembly of the full SNARE complex from the template complex (*Figure 10A*, #4–5; *Figure 10—figure supplements 2* and *3*). Taken together, these results indicate that Munc18-3 and Vps33 catalyze SNARE assembly by templating SNARE folding and association in a manner analogous to that observed for Munc18-1, in strong support of a conserved templating mechanism underlying SM protein function.

## Discussion

Using geometrically faithful single-molecule experiments, we have mapped out a new pathway for the assembly of neuronal SNARE complexes. As suggested previously (*Baker et al., 2015*; *Sitarska et al., 2017*), the key intermediate is a template complex in which the SM protein Munc18-

1 serves as the template to arrange the Qa-SNARE syntaxin and the R-SNARE VAMP2 in a Y-shaped conformation with aligned NTDs and splayed CTDs. Although the first 3–4 layers of the NTDs are not expected to interact directly with the template, they nonetheless appear to be properly zippered in the template complex. Our experiments further indicate that the Qbc-SNARE SNAP-25 binds rapidly to the template complex, presumably by recognizing the properly aligned NTDs of the Qa- and R-SNAREs. Finally, full zippering happens in a single, apparently cooperative transition. Like most enzymatic intermediate states, the template complex is relatively unstable (see Materials and methods for further analysis), preventing it from functioning as a kinetic trap. Nevertheless, our experiments show that it is an obligatory and productive intermediate, which promotes both the speed and the accuracy of SNARE assembly. In addition, the extensive SM-SNARE interactions within the template complex presumably help to prevent the formation of non-cognate SNARE complexes, thereby enhancing the specificity of SNARE pairing (*Shen et al., 2007*).

Our data appear to be inconsistent with an alternative model in which Munc18-1 binds to and activates the t-SNARE complex to promote t-v zippering (*Dawidowski and Cafiso, 2016*; *Jakhanwal et al., 2017*) (*Figure 1*, upper pathway). We found that Munc18-1 inhibited t-SNARE complex formation and minimally affected t-v zippering, consistent with previous reports (*Ma et al., 2013*; *Pobbati et al., 2006*; *Zhang et al., 2015*). Shen et al. found that fusion between t-liposomes (containing syntaxin:SNAP-25 complexes) and v-liposomes (containing VAMP2) was only stimulated by Munc18-1 after all three were preincubated at 4°C for 3 hr (*Shen et al., 2007*). This preincubation, under conditions that prevent fusion, was presumably needed to allow formation of the Munc18-1-stabilized partially-zippered SNARE complexes we previously observed (*Ma et al., 2015*) (*Figure 1*, state v). Preincubating Munc18-1 with t-liposomes alone resulted in little stimulation (*Shen et al., 2007*), inconsistent with the formation of an activated Munc18-1:t-SNARE complex. Moreover, t-SNARE complexes, because they are vulnerable to the ubiquitous SNARE disassembly machinery NSF/SNAP (*Lai et al., 2017*; *Yavuz et al., 2018*), do not appear to represent plausible intermediates in physiological SNARE assembly pathways. Finally, in vivo imaging of SNARE proteins during exocytosis suggested that SNAP-25 is recruited to the fusion site and undergoes a conformational change immediately prior to membrane fusion (*Gandasi and Barg, 2014*; *Zhao et al., 2013*). A stable and pre-formed t-SNARE complex is unlikely to undergo the large SNAP-25 conformational change observed in these experiments (*Zhang et al., 2016b*).

Our results identify a new role for the NRD of syntaxin in stabilizing the template complex. The stabilizing effect of the NRD is partitioned between its N-peptide and its $H_{abc}$ domain (*Figure 2A–B*; *Figure 4*). The stabilizing effect of the N-peptide, which binds to a distal site on Munc18-1 (*Burkhardt et al., 2008*), is unsurprising, as the N-peptide has long been thought to promote interactions between Munc18-1 and partially or completely folded SNARE complexes (*Dulubova et al., 2007*; *Ma et al., 2015*; *Shen et al., 2007*). However, the stabilizing effect of the $H_{abc}$ domain, even when it is added in trans, is unexpected. This role adds to the others that have been ascribed to the syntaxin NRD and that have complicated efforts to elucidate the physiological neuronal SNARE assembly pathway (*Meijer et al., 2012*; *Shen et al., 2010*; *Zhou et al., 2013*). By contrast, the Qa-SNARE Vam3 does not adopt a closed conformation (*Dulubova et al., 2001*), a simplifying feature that prompted us to omit its NRD from both our earlier crystallographic studies (*Baker et al., 2015*) and from the single-molecule experiments reported here. Fortunately, the Vps33 template complex was observable in the absence of the Qa-SNARE NRD (*Figure 10*). Thus, whereas template complexes appear to be a general feature of SM-mediated SNARE assembly, their stabilization via Qa-SNARE NRDs may represent a more specialized elaboration.

Other factors involved in neurotransmitter release may impinge upon the template complex we have identified. For example, Munc13-1 plays important roles in opening syntaxin and promoting proper SNARE complex assembly (*Lai et al., 2017*; *Ma et al., 2011*; *Yang et al., 2015*). The opener function of Munc13-1 was circumvented in our studies by two orthogonal strategies, each of which precludes full syntaxin closure. Notably, however, fully closed syntaxin was only marginally more stable than the template complex ($7.2 \pm 0.2$ $k_BT$ vs $5.2 \pm 0.2$ $k_BT$; *Figure 2—figure supplement 4*). Given that Munc13-1 binds weakly to both syntaxin and VAMP2 at sites likely complementary to those involved in Munc18-1 binding (*Lai et al., 2017*; *Sitarska et al., 2017*; *Wang et al., 2017*), it is attractive to hypothesize that Munc13-1 exerts both its syntaxin opening and SNARE proofreading activities by binding to and stabilizing the template complex (*Figure 1*, from state i to state iv). Additional factors including complexin and synaptotagmin likely capture the SNARE complex

downstream of the template complex, for example by binding to the partially zippered SNARE complex (state v), thereby imposing further regulatory constraints – especially calcium triggering – on synaptic vesicle fusion (*Brunger et al., 2018*).

The finding that several SM proteins – Munc18-1, Munc18-3, and Vps33 – all catalyze SNARE assembly via a template complex confirms that this is a key conserved function of SM proteins. SNARE zippering, because it involves the coupled folding and assembly of four intrinsically disordered SNARE motifs, is inefficient (*Brunger, 2005*; *Lai et al., 2017*). SM proteins, by increasing both the rate and fidelity of SNARE assembly, are likely to be key factors for the control of membrane fusion in vivo. Templated assembly may also resist the disassembly activity of NSF/SNAP.

We propose a working model, using neuronal exocytosis as an example, that places our results in the context of the full fusion machinery (*Figure 1*). First, SNAREs and SM proteins are recruited to, and thereby concentrated at, the future site of membrane fusion during vesicle docking (*Figure 1*, state i). Munc13-1 helps bridge vesicle and plasma membranes and recruit SNAREs, and catalyzes opening of the closed syntaxin (state ii) (*Lai et al., 2017*; *Ma et al., 2011*; *Ma et al., 2013*). Subsequently, Munc18-1 binds to the R-SNARE to form the template complex (iv), which may be further stabilized by Munc13-1. Binding of SNAP-25 generates a partially-zippered SNARE complex stabilized by Munc18-1 (state v). Synaptotagmin and complexin likely associate with the partially zippered SNARE complex, stabilizing it in a primed trans-SNARE complex in preparation for calcium-triggered exocytosis (*Südhof and Rothman, 2009*). Finally, calcium triggers fast CTD zippering and Munc18-1 displacement (state vi), inducing membrane fusion.

## Materials and methods

### Key resources table

| Reagent type (species) or resource | Designation | Source or reference | Identifiers | Additional information |
|---|---|---|---|---|
| Strain, strain background (*species*) | BL21 Gold (DE3) competent cells | Agilent echnologies | Cat#230132 | |
| Commercial assay or kit | BirA-500: BirA biotin-protein ligase standard reaction kit | Avidity | Cat#BirA500 | |
| Chemical compound, drug | dNTP mix (10 mM) | Invitrogen | Cat#18427013 | |
| Chemical compound, drug | 2,2'-dithiodipyridine disulfide (DTDP) | Sigma-Aldrich | CAS#2127-03-9 | |
| Chemical compound, drug | Protease inhibitor cocktail tablet, EDTA free | Roche | Cat#05892791001 | |
| Peptide, recombinant protein | Catalase from bovine liver | Sigma-Aldrich | CAS#9001-05-2 | |
| Peptide, recombinant protein | Glucose Oxidase from *Aspergillus niger* | Sigma-Aldrich | CAS#9001-37-0 | |
| Software, algorithm | LabVIEW VIs | (*Gao et al., 2012*) | | instrument control, data acquisition, and data analysis |
| Software, algorithm | MATLAB scripts | (*Gao et al., 2012*) (*Rebane et al., 2016*) | | data analysis |
| Software, algorithm | Geneious | Geneious | | DNA primer design |
| Software, algorithm | GraphPad Prism7 | GraphPad Software | | |
| Other | Micro Bio-spin six columns | Bio-RAD | Cat#732–6221 | |

*Continued on next page*

*Continued*

| Reagent type (species) or resource | Designation | Source or reference | Identifiers | Additional information |
|---|---|---|---|---|
| Other | Ni Sepharose 6 Fast Flow | GE healthcare Lifesciences | Cat#17531801 | |
| Other | Anti-digoxigenin antibody coated polystyrene particles | Spherotech | Cat#DIGP-20–2 | 2.1 µm, called DIG beads |
| Other | Streptavidin-coated polystyrene particles | Spherotech | Cat#SVP-15–5 | 1.8 µm |
| Other | Customized glass tubing: bead dispenser tubes with 100 µm outer diameter (OD) and 25 µm inner diameter (ID) | King Precision Glass, Inc | | |
| Other | Polyethylene tubing PE50 | Becton Dickinson | Cat# 22–270835 | |
| Other | Dual optical trap setup | (*Gao et al., 2012*) | | |

## SNARE constructs

The cytoplasmic domains of rat neuronal SNAREs, and the SNARE motifs of *C. thermophilum* vacuolar SNAREs, were used. Their sequences are listed below and their domains and crosslinking sites are shown in *Figure 2—figure supplement 1*. In the sequences below, numbers in parenthesis after each construct name indicate the amino acid numbering in the original protein sequence if there is any truncation, followed by the mutated amino acids, if any, which are also colored red in the sequence. The amino acids in the zero layer are colored blue. Extra sequences, including linker sequences, are underlined, with Avi-tags or cysteine residues used for crosslinking shown in bold.

### Vamp2 (1–96, N29C)

MSATAATVPPAAPAGEGGPPAPPPNLTS**C**RRLQQTQAQVDEVVDIMRVNVDKVLE**R**DQKLSELDDRADALQAGASQFETSAAKLKRKYWWKNLKMMGGSGNGSGGL**C**TPSRGGDYKDDDDK

### Syntaxin-1A (1–265, R198C, C145S)

MKDRTQELRTAKDSDDDDDVTVTVDRDRFMDEFFEQVEEIRGFIDKIAENVEEVKRKHSAILASPNPDEKTKEELEELMSDIKKTANKVRSKLKSIEQSIEQEEGLNRSSADLRIRKTQHSTLSRKFVEVMSEYNATQSDYRE**R**SKGRIQRQLEITGRTTTSEELEDMLESGNPAIFASGIIMDSSISKQALSEIET**C**HSEIIKLENSIRELHDMFMDMAMLVES**Q**GEMIDRIEYNVEHAVDYVERAVSDTKKAVKYQSKARRKKGGSGNGGSGS**GLNDIFEAQKIEWHE**

### Syntaxin-1A (1–265, I187C, C145S)

MKDRTQELRTAKDSDDDDDVTVTVDRDRFMDEFFEQVEEIRGFIDKIAENVEEVKRKHSAILASPNPDEKTKEELEELMSDIKKTANKVRSKLKSIEQSIEQEEGLNRSSADLRIRKTQHSTLSRKFVEVMSEYNATQSDYRE**R**SKGRIQRQLEITGRTTTSEELEDMLESGNPAIFASGIIMDSS**C**SKQALSEIETRHSEIIKLENSIRELHDMFMDMAMLVES**Q**GEMIDRIEYNVEHAVDYVERAVSDTKKAVKYQSKARRKKGGSGNGGSGS**GLNDIFEAQKIEWHE**

### Syntaxin-1A, ΔNRD (187–265, R198C)

ISKQALSEIET**C**HSEIIKLENSIRELHDMFMDMAMLVES**Q**GEMIDRIEYNVEHAVDYVERAVSDTKKAVKYQSKARRKKGGSGNGGSGS**GLNDIFEAQKIEWHE**

### Syntaxin-1A, ΔH$_{abc}$ (Δ27–146, R198C)

MKDRTQELRTAKDSDDDDDVTVTVDRTSGRIQRQLEITGRTTTSEELEDMLESGNPAIFASGIIMDSSISKQALSEIET**C**HSEIIKLENSIRELHDMFMDMAMLVES**Q**GEMIDRIEYNVEHAVDYVERAVSDTKKAVKYQSKARRKKGGSGNGGSGS**GLNDIFEAQKIEWHE**

## SNAP-25B (C85S, C88S, C90S, C92S)

MAEDADMRNELEEMQRRADQLADESLESTRRMLQLVEESKDAGIRTLVMLDE*Q*GEQLERIEEGMDQI
NKDMKEAEKNLTDLGKF*S*GL*S*V*S*PSNKLKSSDAYKKAWGNNQDGVVASQPARVVDEREQMAISGGFI
RRVTNDARENEMDENLEQVSGIIGNLRHMALDMGNEIDT*Q*NRQIDRIMEKADSNKTRIDEANQRATK
MLGSG

## Syntaxin-4 (1–273, Q194C)

MRDRTHELRQGDNISDDEDEVRVALVVHSGAARLSSPDDEFFQKVQTIRQTMAKLESKVRELEKQQVTILA
TPLPEESMKQGLQNLREEIKQLGREVRAQLKAIEPQKEEADENYNSVNTRMKKTQHGVLSQQFVELINK
SNSMQSEYREKNVERIRRQLKITNAGMVSDEELEQMLDSGQSEVFVSNILKDT**C**VTRQALNEISARHSEI
QQLERSIRELHEIFTFLATEVEM*Q*GEMINRIEKNILSSADYVERGQEHVKIALENQKKARKKK
<u>GGSGNGGSGS</u>**GLNDIFEAQKIEWHE**

## Syntaxin-4, ΔNRD (191–273, R206C)

GKDTQVTRQALNEISA**C**HSEIQQLERSIRELHEIFTFLATEVEM*Q*GEMINRIEKNILSSADYVERGQEHVKIA
LENQKKARKKK<u>GGSGNGGSGS</u>**GLNDIFEAQKIEWHE**

## SNAP-23 (C79S, C80S, C83S, C85S, C87S)

MDDLSPEEIQLRAHQVTDESLESTRRILGLAIES*Q*DAGIKTITMLDEQGEQLNRIEEGMDQINKDMREAEK
TLTELNK*SS*GL*S*V*S*P*S*NRTKNFESGKNYKATWGDGGDSSPSNVVSKQPSRITNGQPQQTTGAASGGYIKRI
TNDAREDEMEENLTQVGSILGNLKNMALDMGNEIDAQNQQIQKITEKADTNKNRIDIANTRAKKLIDS

## Nyv1 (148-218)

<u>GSS**C**GGG</u>VENNGGDSINSVQREIEDVRGIMSRNIEGLLE**R**GERIDLLVDKTDRLGGSAREFRLRSRGLKRK
MWWKNVK<u>GGSGNGSGGG**C**KAAA</u>

## Vam3 (181-252)

<u>GSS**C**GGG</u>LILEREEEIRNIEQGVSDLNVLFQQVAQLVAE*Q*GEVLDTIERNVEAVGDDTRGADRELRAAA
RYQKRARSRM<u>GGSGNGSGLKNSGGSGSGGNRGGSDSGGSG</u>**GLNDIFEAQKIEWHE**AAA

## Vti1 (126-190)

GSMLDRSTQRLKASQALAAETEAIGASMLAQLQQ*Q*REVIANTTRILYESEGYVDRSIKSLKGIARRM

## Vam7 (308-371)

GSQKLDEQEEYVKDIGVHVRRLRHLGTEIYNAIE*Q*SKDDLDTLDQGLTRLGNGLDKAKALEKKVSGR

## DNA handle preparation

The DNA handle used in our single-molecule experiments is 2,260 bp in length and contains a thiol group (-SH) at one end and two digoxigenin moieties at the other end. The DNA handle was generated by PCR and purified using a PCR purification kit (Qiagen). Both labels were added to the 5′ ends of the PCR primers during synthesis.

## Protein purification

The coding sequences for rat or human syntaxin-1A, VAMP2, Munc18-1, and syntaxin-4 were cloned into pET-SUMO (Invitrogen), which introduced a His$_6$-SUMO tag at the N-termini of the proteins. The coding sequences for rat SNAP-25B and SNAP-23 were cloned into pET-15b (Novagen), which introduced a His$_6$ tag at the N-terminus of the protein. The coding sequence for rat Munc18-3 was cloned into pET-15a (Novagen) and codon-optimized for protein expression in bacteria (*Morey et al., 2017*). The plasmids were transformed into *Escherichia coli* BL21 (DE3) cells (Agilent Technologies), which were then grown in LB media supplemented with the appropriate antibiotics at 37°C until the OD at 600 nm reached 0.6–0.8. The cells were induced with 1 mM IPTG at 37°C for 5 hr. Variants of syntaxin-1A, VAMP2, SNAP-25B and Munc18-1 were prepared using standard PCR-based site-directed mutagenesis (Qiagen).

The neuronal SNARE proteins and Munc18-1 were purified using His-tag affinity purification, as previously described (*Gao et al., 2012*; *Ma et al., 2015*). Briefly, the cells were disrupted in HEPES

buffer (25 mM HEPES, 400 mM KCl, 10% glycerol, 0.5 mM TCEP, pH 7.7) containing 10 mM imidazole and one tablet of EDTA free protease inhibitor cocktail (cOmplete, Roche). Cell lysates were cleared by ultracentrifugation. The resulting supernatant was mixed with Ni-NTA resin overnight, after which the resin was washed successively with HEPES buffer containing 20, 40, and 60 mM imidazole. SNAP-25B, VAMP2 and Munc18-1 were eluted in HEPES buffer containing 300 mM imidazole. Syntaxin-1A was eluted in biotinylation buffer (25 mM HEPES, 200 mM potassium glutamate, 300 mM imidazole, pH 7.7) for future biotinylation (see below). For VAMP2 and Munc18-1, the $His_6$-SUMO tags were cleaved by SUMO proteases at 4°C overnight. The cleaved tags were removed by binding to Ni-NTA resin followed by centrifugation.

The *Chaetomium thermophilum* vacuolar SNARE motifs and Vps33 were purified using a previously described protocol with minor modifications (*Baker et al., 2015*). The *C. thermophilum* SNARE motifs were cloned into a modified pQLinkH vector, resulting in an N-terminal $His_7$-MBP-tag. The plasmids were transformed into *E. coli* C43 (DE3) cells (Lucigen), which were grown in LB media supplemented with ampicillin at 37°C until the OD at 600 nm reached ~0.6. The cells were induced with 0.5 nm IPTG at 30°C for 4 hr and disrupted in lysis buffer (20 mM HEPES, pH 8.0, 350 mM NaCl, 10 mM β-mercaptoethanol and 1 mM PMSF) supplemented with 40 mM imidazole. The lysate was cleared by centrifugation at 17,000 g for 30 min. The $His_7$-MBP-tagged SNARE domains and $His_7$-tagged Vps33 were purified by binding to Ni-NTA resin for several hours, followed by three washes with lysis buffer supplemented with 40 mM imidazole, and elution in lysis buffer supplemented with 300 mM imidazole. For Vps33, the protein was concentrated, followed by size exclusion chromatography on a S200 column equilibrated with gel filtration buffer (20 mM Tris pH 8.0, 250 mM NaCl, 5% glycerol and 0.5 mM TCEP). For the SNARE domains, the $His_7$-MBP tag was removed by incubation with TEV protease with a protein:protease ratio of 20:1 for 3 hr at room temperature. The sample was pre-cleared by running on a gravity flow amylose column and concentrated, followed by size exclusion chromatography on a S75 column equilibrated with gel filtration buffer. Residual $His_7$-MBP was removed using a gravity flow amylose column.

After purification, Qa-SNAREs (syntaxin-1A and Vam3) were biotinylated at the Avi-tag in the presence of 50 μg/mL BirA, 50 mM bicine buffer, pH 8.3, 10 mM ATP, 10 mM magnesium acetate, and 50 μM d-biotin (Avidity) at 4°C overnight (*Gao et al., 2012*; *Jiao et al., 2017*).

## SNARE complex formation

To form synaptic SNARE complexes, syntaxin-1A, SNAP-25B, and VAMP2 were mixed in a molar ratio of 0.8:1:1.2 and incubated at 4°C overnight in the HEPES buffer (pH 7.7) with 2 mM TCEP. The SNARE complexes were purified using the His-tag on SNAP-25B (*Gao et al., 2012*). The quality of the purified neuronal SNARE complex was confirmed by its SDS-resistance in denaturing gel electrophoresis. To form *C. thermophilum* vacuolar SNARE complexes, the His-MBP-tagged Nyv1, Vam3, Vti1 and Vam7, 250 nmol each, were mixed and incubated overnight at 4°C. The complexes were separated from unbound SNAREs by size exclusion chromatography. The His-MBP-tags were cleaved from the SNARE domains using TEV protease and removed by binding to amylose resin. The vacuolar SNARE complex was stored in 20 mM Tris, pH 8.0, 250 mM NaCl, 5% glycerol, 0.5 mM TCEP.

## Crosslinking

After the SNARE complexes were formed, we crosslinked the R and Qa-SNAREs at the N-termini of their SNARE motifs and the R-SNARE C-terminus and the 2,260 bp-DNA handle. To this end, both SNARE complexes and DNA handles were treated with 2 mM TCEP for 1 hr at room temperature, after which Bio-Spin six columns (Bio-Rad) were used to change the buffer to crosslinking buffer A (100 mM phosphate buffer, 500 mM NaCl, pH 5.8) for DNA handles or crosslinking buffer B (100 mM phosphate buffer, 500 mM NaCl, pH 8.5) for SNARE complexes. Next, DNA handles were incubated with 1 mM 2,2'-dithiodipyridine disulfide (DTDP) for 1 hr at room temperature to activate the thiol group for the following crosslinking reaction. After incubation, the DNA handle was purified using a PCR purification kit and eluted in crosslinking buffer B to remove excess DTDP. Finally, the SNARE complexes were mixed with the DTDP-treated DNA handles in a 50:1 molar ratio in crosslinking buffer B and incubated at room temperature overnight, as previously described (*Gao et al., 2012*).

## Single-molecule manipulation experiments

All pulling experiments were performed using dual-trap high-resolution optical tweezers as previously described (*Gao et al., 2012*; *Ma et al., 2019*; *Ma et al., 2015*). Briefly, an aliquot of the cross-linked protein-DNA mixture containing 10–100 ng DNA was mixed with 10 µL 2.1 µm diameter anti-digoxigenin antibody coated polystyrene beads (Spherotech) and incubated at room temperature for 15 min. Then the anti-digoxigenin coated beads and 2 µL 1.7 µm diameter streptavidin-coated beads (Spherotech) were diluted in 1 mL PBS buffer (137 mM NaCl, 2.7 mM KCl, 8.1 mM $Na_2HPO_4$, 1.8 mM $KH_2PO_4$, pH 7.4). Subsequently, the bead solutions were separately injected into the top and bottom channels of a homemade microfluidic chamber as described below. The central channel contained PBS buffer with an oxygen scavenging system comprising 400 mg/mL glucose (Sigma-Aldrich), 0.02 unit/mL glucose oxidase (Sigma-Aldrich), and 0.06 unit/mL catalase (Sigma-Aldrich). A single anti-digoxigenin-coated bead was trapped and brought close to a single streptavidin-coated bead held in another optical trap to form a single SNARE-DNA tether between the two beads.

A single SNARE protein (Qa), SNARE conjugate (Qa-R), SNARE complex, or SNARE/SM complex (collectively called the protein complex below) was pulled or relaxed by moving one of the optical traps at a speed of 10 nm/s. In a typical single-molecule manipulation experiment, a single protein complex was first pulled to a high force to completely disassemble the complex, yielding information on the stability and structure of the complex. Then the complex was relaxed to observe its possible refolding or re-assembly. To better observe the assembly of the template complex or the SNARE complex, during relaxation the protein complex was often held at constant trap separations in a force range of 2–8 pN for various times. The formation probability of the complex generally increased as the waiting time increased. Therefore, the formation probability of the template complex or the SNARE four-helix bundle reported in the main text, including *Table 1*, was determined with a maximum waiting time of 1 min.

## Dual-trap high-resolution optical tweezers

The optical tweezers used in our experiments are home-built and described in detail elsewhere (*Gao et al., 2012*; *Ma et al., 2015*; *Sirinakis et al., 2012*). Briefly, the tweezers are assembled on an optical table located in an acoustically isolated, temperature- and air-flow-controlled room. A 1064 nm laser beam from a 4 W Nd:YVO4 diode pumped solid state laser (Spectr-Physics, CA) is expanded by a telescope by about five fold, and split by a polarizing beam splitter (PBS) into two orthogonally polarized laser beams. The two beams are reflected by two mirrors and combined by another PBS. One of the mirrors is mounted on a nano-positioning stage that can tip/tilt in two axes with high resolution (Mad City Labs, WI). The combined beams are further expanded by about two-fold and collimated by another telescope, and focused by a water immersion 60X objective with a numerical aperture of 1.2 (Olympus, PA), forming two optical traps in a central channel of the microfluidic chamber. One of the optical traps can be moved in the sample plane with sub-angstrom resolution via the nano-positioning stage. The flow cell is formed between two coverslips sandwiched by Parafilm cut into three parallel channels. The top and bottom channels are connected to the central channel by glass tubing. The outgoing laser beams are collected and collimated by an identical objective, split again by a PBS, and projected onto two position-sensitive detectors (Pacific Silicon Sensor, CA), which detect displacements of the two beads in optical traps through back-focal-plane interferometry. The optical tweezers are calibrated before each single-molecule experiment by measuring the Brownian motion of the trapped beads, which yields the power-spectrum density distributions of bead displacements. The force constants of optical traps are determined by fitting the measured power-spectrum density distributions with a Lorentzian function.

## Circular dichroism (CD) spectra of Munc18-1

CD spectra of WT and mutant Munc18-1 proteins were measured in 20 mM phosphate buffer using an Applied Photophysics Chirascan equipped with a 2 mm quartz cell. The readings were made at 1 nm intervals, and each data point represents an average of 6 scans at a speed of 120 nm/min over the wavelength range of 190 to 250 nm.

## Derivations of protein unfolding energy and folding and unfolding rates from force-dependent measurements

Our methods of data analysis and the relevant Matlab codes are described in detail elsewhere (*Gao et al., 2012*; *Rebane et al., 2016*; *Zhang et al., 2016a*). Briefly, the extension-time trajectories obtained at constant trap separations or mean forces were first analyzed by two-state hidden-Markov modeling (*McKinney et al., 2006*; *Zhang et al., 2016a*), which revealed the idealized state transitions, extension changes, unfolding probabilities, and folding and unfolding rates. The MATLAB codes for hidden-Markov modeling can be found in Source code file 1. These measurements were used to derive the folding intermediates and their associated energy and kinetics.

We quantified the structural change of a single protein based on the measured force and extensions (*Rebane et al., 2016*). The control parameter of our pulling experiment is the separation between two optical traps ($D$). Given the trap separation, the extension ($X$) and tension ($F$) of the protein-DNA tether are calculated as

$$X = D - x_1 - x_2 \tag{2}$$

$$F = (F_1 + F_2)/2, \tag{3}$$

respectively, where $x_1$ and $x_2$ are displacements of the two beads in optical traps, and $F_1$ and $F_2$ are the corresponding forces applied to the beads. Both bead displacement $x$ and the force $F$ are derived from voltage outputs of the position-sensitive detectors after proper calibrations (*Sirinakis et al., 2012*). In *Equation 2*, we have defined a default relative trap separation by neglecting the contribution of constant bead diameters. It is this relative trap separation that is shown in *Figure 2—figure supplement 2*. As a protein molecule unfolds, its extension ($x_m$) increases, which leads to retraction of both beads in their optical traps and the accompanying decrease in tension (*Figure 2—figure supplement 2*). Thus, during protein folding and unfolding transitions, the tether tension changes in an out-of-phase manner with respect to the tether extension, and thus is state-dependent. In the constant trap separation, the mean force is defined as the mean of the average forces associated with the folded and unfolded states (*Rebane et al., 2016*).

We modeled the unfolded peptide and the DNA handle by a worm-like chain model (*Marko and Siggia, 1995*). Based on this model, the stretching force $F$ and the entropic energy $E$ of a semi-flexible polymer chain are related to its extension $x$, contour length $L$, and persistence length $P$ by the follow formulae

$$F = \frac{k_B T}{P} \left[ \frac{1}{4\left(1 - \frac{x}{L}\right)^2} + \frac{x}{L} - \frac{1}{4} \right] \tag{4}$$

$$E = \frac{k_B T}{P} \frac{L}{4\left(1 - \frac{x}{L}\right)} \left[ 3\left(\frac{x}{L}\right)^2 - 2\left(\frac{x}{L}\right)^3 \right], \tag{5}$$

respectively. We adopted a persistence length 40 nm for DNA and 0.6 nm for the unfolded polypeptide (*Gao et al., 2012*; *Ma et al., 2015*; *Rebane et al., 2016*). Because the DNA extension ($x_{DNA}$) is known given a force or trap separation via *Equation 4*, the extension of the protein can be calculated as $x_m = X - x_{DNA}$. The protein extension generally comprises the extensions of the unfolded polypeptide portion ($x_p$) and the folded portion ($H$) of the protein if any, or $x_m = x_p + H$. The former can be again calculated by *Equation 4*, given the contour length of the unfolded polypeptide ($L_p$), whereas the latter can be treated as a force-independent constant, or a hard core of the protein (*Rebane et al., 2016*). Here, the size of the hard core is determined from the two pulling sites on the folded protein portion, which changes with the protein state. Thus, the contour length of the unfolded polypeptide and the size of the folded protein portion are correlated and can be determined based on a structural model for protein transitions. To derive the structure of the template complex, we assumed a change in the hard core size of 3 nm for the transition from the partially closed syntaxin to the folded template complex, as determined from the structure-based model (*Baker et al., 2015*). Consequently, we could determine the contour length of the polypeptide chain in the Qa-R conjugate that is either free or bound by Munc18-1. The number of amino acids in a

polypeptide is its contour length divided by the contour length per amino acid, which is chosen to be 0.365 nm (*Gao et al., 2012*; *Rebane et al., 2016*). The number of amino acids in the completely unfolded SNARE state 5 is known (*Figure 2—figure supplement 1*), which helps derive the structure of the template complex based on the extension change during the template complex transition. We determined that 87 (±2, S.D.) amino acids are sequestered in the folded template complex, including the N-terminal loop formed between syntaxin-1 and VAMP2 due to crosslinking. Based on our construct design (*Figure 2—figure supplement 1*), this length is consistent with the structure of the predicted template complex (*Figure 1A*).

Similarly, we modeled the total free energy of the whole dumb-bell system in optical traps, or

$$G = \frac{F^2}{2k_{traps}} + E_{DNA} + E_p + V,\qquad(6)$$

where the first term represents the potential energy of the two beads in optical traps with $k_{traps} = k_1 k_2/(k_1 + k_2)$ the effective force constant of the two traps, the second and third terms are entropic energies of the DNA handle and the unfolded polypeptide, respectively, calculated by *Equation 5*, and the last term is the free energy of the protein at zero force. Based on the Boltzmann distribution, the protein unfolding energy $\Delta V$ can be determined by fitting the measured unfolding probability using *Equation 6*. *Equation 6* can be similarly applied to the transition state of protein folding (*Rebane et al., 2016*). With Kramers' rate equation, the folding and unfolding rates are calculated. By fitting the calculated rates to the measured rates, we derive the energy and conformation of the transition state, which also yield the folding and unfolding rates at zero force. Complete data sets from individual molecules are separately fit and the unfolding energies and transition rates, typically averaged over more than three different molecules, are reported (*Table 1*). The average folding rates and unfolding rates of the WT and mutant template complexes fall in the ranges of 17–568 s$^{-1}$ and 0.1–10 s$^{-1}$, respectively, with a standard error typically close to the corresponding average rate for each template complex.

## Estimation of the affinity between VAMP2 and Munc18-bound syntaxin in the absence of crosslinking

The N-terminal crosslinking between Qa- and R-SNAREs used in our assay is crucial for us to observe and characterize the template complex. The crosslinking destabilizes the closed syntaxin, thereby bypassing the requirement for Munc13-1; mitigates SNARE misassembly, for example, formation of various anti-parallel SNARE bundles (*Lai et al., 2017*); and avoids nonspecific VAMP2-Munc18-1 interactions (*Sitarska et al., 2017*). Therefore, the crosslinking simplifies our experimental design and data interpretation. Three lines of evidence suggest that the crosslinking does not compromise the major conclusions derived from our assay. First, the stability (5.2 ± 0.1 k$_B$T vs 4.8 ± 0.3 k$_B$T) and lifetime (1.4 s vs 0.8 s) of the template complex do not significantly depend upon the crosslinking site used in our assay (at R198C vs I187C, *Figure 2—figure supplements 1* and *7*). This observation suggests that the crosslinking does not alter the structure of the template complex and is likely located at a disordered region. The approximate independence of the folding energy and lifetime on the crosslinking site has been extensively tested in other systems, such as DNA hairpins (*Woodside et al., 2006*) and protein coiled coils (*Jiao et al., 2015*). Second, the derived template model recapitulates many distinct features of the fusion machinery, including its dependence upon NRD, phosphorylation, and various mutations. Finally, the crosslinking increases the local SNARE concentration around Munc18-1, which mimics the environment of SNARE assembly and membrane fusion in vivo due to vesicle tethering and SNARE recruitment. For example, Munc13-1 essentially crosslinks both syntaxin and VAMP2 by simultaneously binding the two (*Sitarska et al., 2017*; *Wang et al., 2017*) (*Figure 1*).

The effective concentration due to the crosslinking can be quantified, which is used to estimate the binding affinity between VAMP2 and partially closed syntaxin bound by Munc18-1 in the absence of crosslinking (*Zhang et al., 2016b*). To derive the local concentration of the crosslinked VAMP2 (at R198C) around the partially closed syntaxin bound by Munc18-1, we made three assumptions: 1) the partially closed syntaxin bound by Munc18-1 has a conformation similar to the conformation seen in the crystal structure of the Vps33:Vam3 complex (*Baker et al., 2015*); 2) The kinetics of VAMP2 binding and unbinding are dominated by Phe 77 insertion into and detachment from the F-binding

pocket on the Munc18-1 surface, respectively, as is supported by our data; and 3) the unfolded SNARE polypeptides are described by a Gaussian chain model. Thus, VAMP2 binding to the partially closed syntaxin bound by Munc18-1 can be modeled by Phe 77 binding to the F-binding pocket while Phe 77 is tethered to the $-7$ layer of Vam3 (i.e. the N-terminus of the main Vam3 helix in the Vps33:Vam3 structure, PDB code 5BUZ) via a polypeptide linker. The linker length is 54 amino acids (or $L$ = 19.7 nm in terms of the contour length) for the Qa-R conjugate crosslinked at R198C. The distance between the F-binding pocket and the tethering point ($R$) is measured to be 6.34 nm. Therefore, the effective concentration $c$ of the tethered Phe 77 around its binding pocket is calculated as $c = 3.7 \times 10^{-4}$ M, using the following formula

$$c = \frac{1}{N_A} \left( \frac{3}{4\pi PL} \right)^{\frac{3}{2}} exp \left( -\frac{3R^2}{4PL} \right),\qquad(7)$$

where $N_A$ is Avogadro's number and p=0.6 nm is the persistence length of the polypeptide. Due to coupled binding and folding, the folding and unfolding rates of the template complex we measured should be equal to the binding and dissociation rates estimated here. Therefore, the folding rate $k_f = k_{on} \times$ c, where $k_{on}$ is the bimolecular rate constant for VAMP2 binding to the partially-closed syntaxin. Using our measured folding and unfolding rates for the R198C conjugate, we calculated the VAMP2 binding rate constant as $k_{on} = 3.5 \times 10^5$ s$^{-1}$M$^{-1}$, energy as 13.1 k$_B$T, or a VAMP2 dissociation constant as 2 μM. Supposing that VAMP2 binding to the fully closed syntaxin requires an energy gain of 4.6 k$_B$T compared to the partially closed syntaxin (corresponding to the energy difference between the two syntaxin states), the VAMP2 binding affinity to the fully closed syntaxin is estimated to be 200 μM. This estimation justifies a weak template complex that is difficult to detect in vitro using conventional experimental approaches, but can be observed using our single-molecule approach in combination with N-terminal SNARE crosslinking.

The calculated local concentration of VAMP2 Phe 77 around the F-binding pocket for the Qa-R conjugate crosslinked at I187C ($c = 4.3 \times 10^{-4}$ M) is slightly higher than that for the R198C conjugate ($c = 3.7 \times 10^{-4}$ M). Correspondingly, the unfolding energy of the template complex for the I187C conjugate is expected to be slightly higher by 0.15 k$_B$T, given their equal unbinding rate based on our model. Therefore, our simple molecular model for the template complex transition recapitulates our experimental observations that the stability and lifetime of the template complex is insensitive to the crosslinking site used in our assay.

## Data and software availability
Data and software can be found in the Source data and the Source code file, respectively.

## Acknowledgements
We thank J Rothman and A Horwich for discussion, G Shimamura for technical assistance, and D Fasshauer for providing the plasmid for Munc18-3 purification. This work was supported by NIH grants R01GM093341 and R01GM120193 to YZ, T32GM007223 to JJ, and R01GM071574 to FMH, and by the German Research Foundation (DFG) grant PO2195/1-1 to SAP.

# Additional information

### Funding

| Funder | Grant reference number | Author |
| --- | --- | --- |
| National Institute of General Medical Sciences | R01GM093341 | Yongli Zhang |
| National Institute of General Medical Sciences | R01GM120193 | Yongli Zhang |
| National Institute of General Medical Sciences | R01GM071574 | Frederick M Hughson |
| National Institute of General Medical Sciences | T32GM007223 | Junyi Jiao |

| Deutsche Forschungsge-meinschaft | PO2195/1-1 | Sarah A Port |

The funders had no role in study design, data collection and interpretation, or the decision to submit the work for publication.

### Author contributions

Junyi Jiao, Conceptualization, Data curation, Software, Formal analysis, Validation, Investigation, Visualization, Methodology, Writing—original draft, Writing—review and editing; Mengze He, Conceptualization, Data curation, Formal analysis, Validation, Investigation, Visualization, Methodology, Writing—review and editing; Sarah A Port, Conceptualization, Data curation, Funding acquisition, Investigation, Methodology, Writing—review and editing; Richard W Baker, Conceptualization, Data curation, Investigation, Methodology; Yonggang Xu, Hong Qu, Yujian Xiong, Yukun Wang, Huaizhou Jin, Data curation, Investigation; Travis J Eisemann, Visualization; Frederick M Hughson, Conceptualization, Resources, Supervision, Funding acquisition, Validation, Visualization, Methodology, Writing—original draft, Project administration, Writing—review and editing; Yongli Zhang, Conceptualization, Resources, Data curation, Software, Formal analysis, Supervision, Funding acquisition, Validation, Investigation, Visualization, Methodology, Writing—original draft, Project administration, Writing—review and editing

### Author ORCIDs

Sarah A Port ⓘ http://orcid.org/0000-0002-1897-0510
Frederick M Hughson ⓘ http://orcid.org/0000-0002-4057-0281
Yongli Zhang ⓘ http://orcid.org/0000-0001-7079-7973

### Decision letter and Author response

Decision letter https://doi.org/10.7554/eLife.41771.052
Author response https://doi.org/10.7554/eLife.41771.053

## Additional files

### Supplementary files

• Transparent reporting form
DOI: https://doi.org/10.7554/eLife.41771.050

### Data availability

All data generated or analyzed during this study are included in the manuscript and supporting files. Source data files have been provided for Figures 2-10.

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
