## [Decision Letter]

Thank you for submitting your article "Munc18-1 catalyzes neuronal SNARE assembly by templating SNARE association" for consideration by *eLife*. Your article has been reviewed by three peer reviewers, one of whom, Josep Rizo, has acted as Guest Reviewing Editor. The other two reviewers have agreed to reveal their identity: Tae-Young Yoon (reviewer #2) and Thomas Sollner (reviewer #3). The evaluation has been overseen by the Reviewing Editor and by Vivek Maholtra as the Senior Editor.

The reviewers have discussed the reviews with one another and the Reviewing Editor has drafted this decision to help you prepare a revised submission.

Summary:

This paper is a tour de force of measurements with optical tweezers to study how Munc18-1 influences the assembly of neuronal SNARE complex formed by VAMP2, SNAP-25 and syntaxin-1 (referred to as syntaxin in the manuscript). The study was designed to test a model whereby Munc18-1 templates SNARE complex assembly through interactions with syntaxin and VAMP2. This model emerged from tantalizing evidence provided by two crystal structures of Vps33, the yeast vacuolar homologue of Munc18-1, bound to either Vam3 (syntaxin homologue) or Nyv1 (VAMP2 homologue), which showed that simultaneous binding of Vam3 and Nyv1 to Vps33 would place the two SNAREs in close proximity and in register to form the SNARE complex. Multiple results from other studies had suggested that Munc18-1 interacts with syntaxin and VAMP2 in a similar fashion to template SNARE complex assembly, but the validity the model remained to be proven.

Jiao et al. now provide compelling evidence supporting this model by showing how inclusion of Munc18-1 induces formation of intermediate states in force-extension curves of individual complexes where syntaxin and VAMP2 were tethered through covalent links between the N-termini of their SNARE motifs. This intermediate state can be assigned to specific structural models based on the crystal structures of the SNARE complex, of the binary complex formed by Munc18-1 and syntaxin folded into a so-called closed conformation, and of the aforementioned Vps33/Vam3 and Vps33/Nyv1 complexes. The interpretation of the results in terms of the template model is strongly supported by the effects of a large number of mutations in Munc18-1, VAMP2 and syntaxin on the force extension curves, as well as by the correlations between these effects and functional data obtained in previous studies with liposome fusion assays and electrophysiological experiments in neurons. Interestingly, an interaction of Munc18-1 with the N-terminal regulatory domain (NRD) of syntaxin-1 plays a stabilizing role, even when the NRD is provided in trans. Impressively, the authors extend their studies to the Munc18-3:syntaxin-4:SNAP-23:VAMP2 complex, mediating the release of GLUT4-containing vesicles and to the Vps33:Vam3: Nyv1:Vti1:Vam7 complex, mediating vacuolar fusion. Overall, the results demonstrate that SM proteins promote the rate and fidelity of SNARE complex assembly by providing an initial template for Qa and R-SNARE assembly. The reviewers believe that this paper will represent a landmark contribution to the field and strongly support publication.

Essential revisions:

1) There are some limitations to the study that need to be emphasized, and some of the conclusions need to be toned down. In particular, while the existence of the key intermediate that reflects the template function (state 7 in Figure 2D) is well supported by the large number of mutations and the availability of the crystal structures (i.e. the data make sense!), some of the other intermediates are less clear. Two points are particularly unclear. First, intermediate 8 in Figure 2D assumes that SNAP-25 binds to the Munc18-1-syntaxin-VAMP2 complex (firmly concluded in subsection “Munc18-1 stabilizes the SNARE complex in a partially-zippered state”), but the nature of this potential intermediate is highly unclear, as the residues of VAMP2 and syntaxin that are expected to bind to SNAP-25 based on the structure of the SNARE complex are expected to be at least partially buried due to their interactions with Munc18-1 in the template complex. This is a major conundrum that remains to be solved and it should be presented as such. Second, the release of Munc18-1 assumed to occur upon full SNARE complex zippering (transition 9 to 1 in Figure 2D) is prominently mentioned in the abstract and the very last sentence of the paper, but it seems that the data supporting this conclusion are rather indirect. Even if the data were more convincing, interactions of Munc18-1 with the assembled SNARE complex may be stabilized by the membranes (Shen et al., 2007), which are not present in the optical tweezer experiments. Moreover, can the authors exclude that Munc18 remains bound to the syntaxin-1 NRD? Overall, the authors should be very careful in distinguishing throughout the manuscript between conclusions that are well established by the data and others that are more tentative.

2) Some of the effects summarized in Figure 4 are quite small and, while they point in the right direction, their limited magnitude seems a little surprising given the nature of the mutation and/or the previously observed functional effects. One example is the P335A mutation, which has strong functional effects but stabilizes the template complex rather moderately. The authors address this concern in part, but to what extent the small error bars shown in Figure 4 represent the true uncertainty of the effects of the mutations, as error bars derived from fitting data can often be much smaller than the actual uncertainty of the parameter derived from the measurements. In this context, it would be helpful to know how many force-extension curves contribute to the calculations listed for each entry in Table 1. The N values appear to refer to the number of transitions observed, but how many curves were recorded? And if a parameter like the unfolding energy is calculated separately from data from different curves, is the error different than that described in the table? The authors should also provide an indication of statistical significance for selected cases where the effects are small and can be compared with available functional data. Moreover, the very nature of the experiments, which introduced a covalent linkers to tether syntaxin and VAMP2, and to hinder the syntaxin closed conformation, is expected to have a strong effect on folding rates and may influence the effects of the mutations on the 6 to 7 transition for instance, perhaps explaining the modest nature of some effects mentioned above. These limitations should be emphasized and discussed.

3) The most interesting data in this manuscript is probably the observation of the template complex, in which syntaxin and synaptobrevin are held by Munc18-1 in a half-zippered-like state. This template complex is distinguished from the previous model that assumes that the t-SNARE pre-complex is a central, obligatory intermediate in SNARE complex formation. However, it is not clear in the current manuscript how the authors distinguished the template complex from the t-SNARE precomplex in their essentially one-dimensional measurement. For example, in Figure 5B, it was difficult to understand how the authors distinguished spontaneous assembly (i.e., formation of t-SNARE precomplex) from the chaperoned assembly via the template complex. It appears that the template complex occurred at relatively higher tension levels (5 to 10 pN; Figure 2C, Figure 3 and Figure 5D) while the formation of t-SNARE pre-complexes mainly occurred under low pN tension (2 to 4 pN; Figure 5C, but there is also an exception in Figure 5C #1). The authors should provide detailed accounts on how the one-dimensional extension levels (or states) observed for the template complex can be distinguished from those observed for the t-SNARE pre-complexes.

4) In the model for template complex formation (in Figure 2D), State 6 and State 7 are to be characterized by formation of alpha-helical structure in the SNARE motifs of syntaxin and synaptobrevin, respectively. Is there any orthogonal experimental data supporting this notion in addition to the small changes in the extension value of the tweezed SNARE proteins? Although it is not absolutely required for publication of this manuscript, such data would corroborate the authors' claim.

5) Some of the figure legends need a lot more detail to understand what is being shown (e.g. Figure 2C and 5A). For instance, if the first curve was recorded without Munc18-1 (as suggested by the label in the figure) but the others included Munc18-1, this should be explicitly explained in the legend.

---

## [Author Response]

Essential revisions:1) There are some limitations to the study that need to be emphasized, and some of the conclusions need to be toned down. In particular, while the existence of the key intermediate that reflects the template function (state 7 in Figure 2D) is well supported by the large number of mutations and the availability of the crystal structures (i.e. the data make sense!), some of the other intermediates are less clear. Two points are particularly unclear. First, intermediate 8 in Figure 2D assumes that SNAP-25 binds to the Munc18-1-syntaxin-VAMP2 complex (firmly concluded in subsection “Munc18-1 stabilizes the SNARE complex in a partially-zippered state”), but the nature of this potential intermediate is highly unclear, as the residues of VAMP2 and syntaxin that are expected to bind to SNAP-25 based on the structure of the SNARE complex are expected to be at least partially buried due to their interactions with Munc18-1 in the template complex. This is a major conundrum that remains to be solved and it should be presented as such. Second, the release of Munc18-1 assumed to occur upon full SNARE complex zippering (transition 9 to 1 in Figure 2D) is prominently mentioned in the abstract and the very last sentence of the paper, but it seems that the data supporting this conclusion are rather indirect. Even if the data were more convincing, interactions of Munc18-1 with the assembled SNARE complex may be stabilized by the membranes (Shen et al., 2007), which are not present in the optical tweezer experiments. Moreover, can the authors exclude that Munc18 remains bound to the syntaxin-1 NRD? Overall, the authors should be very careful in distinguishing throughout the manuscript between conclusions that are well established by the data and others that are more tentative.

We agree with reviewers that more evidence is required to support our interpretation regarding the Munc18-1-bound partially zippered state and the Munc18-1 state after full SNARE zippering. Consequently, we have decided not to describe the Munc18-1-bound partially zippered state as part of our results and to remove the relevant description in the text and diagram in Figure 2D. Our results reveal that the SNARE complexes formed in the presence of Munc18-1 generally exhibit the identical stepwise unfolding as the purified SNARE complexes. However, our results do not rule out the possibility that after SNARE assembly Munc18-1 remains bound to the SNARE complex in a manner that does not interfere with its unfolding. Therefore, we have removed our statements that Munc18-1 is displaced from the SNARE complex after full SNARE assembly. In subsection “Template complex facilitates SNARE assembly”, we have added:

“14% of the time, however, the subsequent pulling FEC revealed unfolding of the VAMP2 CTD at unusually low force of 4-14 pN (Figure 7A, #2). These are not mis-assembly events, because the SNARE complexes show stepwise NTD unfolding and t-SNARE unfolding identical to the properly assembled SNARE complexes. Further work will be required to determine whether or not these complexes represent on-pathway intermediates.”

2) Some of the effects summarized in Figure 4 are quite small and, while they point in the right direction, their limited magnitude seems a little surprising given the nature of the mutation and/or the previously observed functional effects. One example is the P335A mutation, which has strong functional effects but stabilizes the template complex rather moderately. The authors address this concern in part, but to what extent the small error bars shown in Figure 4 represent the true uncertainty of the effects of the mutations, as error bars derived from fitting data can often be much smaller than the actual uncertainty of the parameter derived from the measurements. In this context, it would be helpful to know how many force-extension curves contribute to the calculations listed for each entry in Table 1. The N values appear to refer to the number of transitions observed, but how many curves were recorded? And if a parameter like the unfolding energy is calculated separately from data from different curves, is the error different than that described in the table? The authors should also provide an indication of statistical significance for selected cases where the effects are small and can be compared with available functional data. Moreover, the very nature of the experiments, which introduced a covalent linkers to tether syntaxin and VAMP2, and to hinder the syntaxin closed conformation, is expected to have a strong effect on folding rates and may influence the effects of the mutations on the 6 to 7 transition for instance, perhaps explaining the modest nature of some effects mentioned above. These limitations should be emphasized and discussed.

In our revision, we have quantified the impact of the template complex stability on membrane fusion and exocytosis, assuming that formation of the template complex is a rate-limiting step of overall SNARE assembly and membrane fusion. Our calculations show that it is possible to explain the strong functional effects of many Munc18-1 mutants by the modest change in the unfolding energy of the template complex measured by us (Figure 4B). This is because the overall rate of membrane fusion is an exponential function of the unfolding energy of the template complex (see Equation 1).

In Table 1, the N values listed in the “template formation” column are the total numbers of pulling and relaxation FECs in which template complex formation is scored. The N values listed in the “SNAP-25B binding” column are the total numbers of relaxation FECs containing template complex transitions in which SNARE-25B binding and SNARE assembly events are scored. We have added footnotes to clarify these N values. For each FEC containing the template complex transition, we measured an equilibrium force and the corresponding extension change. The average equilibrium force listed in Table 1 is the mean of all such forces. To determine the unfolding energy, we first measured template complex transitions at a range of constant forces and determined the equilibrium force, unfolding probability, transition rates, and extension changes (as shown in Figure 3C). Curves of these parameters as a function of force obtained from different Qa-R SNARE conjugates often show systematic shift along the x axis, partially due to the systematic error in absolute force measurement among different molecules (Moffitt et al., 2006). To combine these data obtained from 2-10 Qa-R conjugates, we shifted these curves along the force axis (or x-axis) such that the equilibrium force of each molecule is equal to the average equilibrium force measured from a large number of FECs (shown in Table 1), as is previously described (Ma et al., 2017). Then we nonlinearly fit all these curves simultaneously to determine the unfolding energy and transition rates of the template complex at zero force (Rebane et al., 2016). The resultant unfolding energy is typically more accurate than the unfolding energy determined by measurements from a single molecule. Note that the errors shown throughout our text are the standard error of the mean.

We agree with reviewers that the folding rate of the template complex measured by us depends on the N-terminal crosslinking site used in our experiments, which is also confirmed by our results (compare Figure 3A and Figure 3H). However, the unfolding rate and unfolding energy of the template complex at zero force are insensitive to the N-terminal crosslinking site as long as the site is located in a disordered region away from the structured portion of the template complex, as is confirmed by our experiments. The approximate independence of the folding energy on the crosslinking site has been extensively tested in other systems, such as DNA hairpins (Woodside et al., 2006) and protein coiled coils (Jiao et al., 2015). Therefore, we believe the modest energy difference among various mutants tested by us is not caused by the specific N-terminal crosslinking used in our experiments.

3) The most interesting data in this manuscript is probably the observation of the template complex, in which syntaxin and synaptobrevin are held by Munc18-1 in a half-zippered-like state. This template complex is distinguished from the previous model that assumes that the t-SNARE pre-complex is a central, obligatory intermediate in SNARE complex formation. However, it is not clear in the current manuscript how the authors distinguished the template complex from the t-SNARE precomplex in their essentially one-dimensional measurement. For example, in Figure 5B, it was difficult to understand how the authors distinguished spontaneous assembly (i.e., formation of t-SNARE precomplex) from the chaperoned assembly via the template complex. It appears that the template complex occurred at relatively higher tension levels (5 to 10 pN; Figure 2C, Figure 3 and Figure 5D) while the formation of t-SNARE pre-complexes mainly occurred under low pN tension (2 to 4 pN; Figure 5C, but there is also an exception in Figure 5C #1). The authors should provide detailed accounts on how the one-dimensional extension levels (or states) observed for the template complex can be distinguished from those observed for the t-SNARE pre-complexes.

The two pathways of SNARE assembly, the t-SNARE pathway and the template complex pathway, are distinguished by their distinct folding kinetics and force range seen in the relaxation FECs and the extension-time trajectories at constant force. SNARE assembly via the t-SNARE pathway causes a single extension drop at 2-4 pN, whereas Munc18-1-chaperoned SNARE assembly is characterized by reversible extension flickering in the force range of 3-10 pN followed by an irreversible average extension drop of 5.5 nm. Note that the SNARE assembly event at ~8 pN shown in our original Figure 5C, #1 occurred with a preformed t-SNARE (from state 3), which is different from SNARE assembly starting from de novo t-SNARE assembly (from state 4 or 5; Figure 5C, #4; Figure 5D, a).

To clarify our description on SNARE complex assembly, we have reorganized the original Figure 5 into three separate figures (new Figures 5-7). In addition, we have added the following sentences in the section “Template complex facilitates SNARE assembly”:

“The probability of SNARE assembly per relaxation was measured to assess the efficiency of SNARE assembly. Properly assembled SNARE complexes were taken to be those that: (i) displayed the same extension at low force as the initial purified complex, and (ii) exhibited stepwise unfolding in subsequent rounds of pulling.”

“This ‘spontaneous’ (i.e., Munc18-1-independent) SNARE assembly is observed as a single extension drop in the force range of 2-4 pN (Figure 5C). The t-SNARE complex forms in the same force range (Zhang et al., 2016), indicating that the spontaneous SNARE assembly we observe here is likely limited by t-SNARE formation. Consistent with this interpretation, zippering between the t-SNARE complex and v-SNARE (or t-v zippering) occurs at much higher force (Figure 5B, #1).”

“In a representative experiment, a single Qa-R SNARE conjugate assembled into a proper SNARE complex during each of five consecutive rounds of pulling and relaxation (Figure 7A, #1-#5). Overall, based on 67 pulling and relaxation FECs conducted using 15 Qa-R SNARE conjugates, proper assembly was observed with a probability of 0.53 per relaxation (Figure 6). However, SNARE assembly tended to occur consecutively: the conditional probability to observe one SNARE assembly event after another event was 0.79 (N=52) (Figure 7A, #1-#5), likely mediated by a single Munc18-1 molecule.”

We have also added a sentence in the legend for Figure 5B:

“In FEC #1, the SNARE complex was relaxed just after the t- and v-SNAREs were unzipped to observe t-v zippering at low force (blue arrow).”

4) In the model for template complex formation (in Figure 2D), State 6 and State 7 are to be characterized by formation of alpha-helical structure in the SNARE motifs of syntaxin and synaptobrevin, respectively. Is there any orthogonal experimental data supporting this notion in addition to the small changes in the extension value of the tweezed SNARE proteins? Although it is not absolutely required for publication of this manuscript, such data would corroborate the authors' claim.

In State 6 and State 7, the alpha-helical structures in the SNARE motifs of syntaxin and synaptobrevin are derived from the corresponding structures of closed syntaxin (PDB code 3C98) and the structural model of the template complex, respectively; the latter is inferred from our previous crystal structures of SM:QaSNARE (5BUZ) and SM:R-SNARE (5BV0) complexes. Unfortunately, it is difficult if not impossible to confirm the helical structures using our single-molecule approaches. In the future, however, we hope to use X-ray crystallography to provide an atomic-resolution view of the Munc18:syntaxin:synaptobrevin template complex.

5) Some of the figure legends need a lot more detail to understand what is being shown (e.g. Figure 2C and 5A). For instance, if the first curve was recorded without Munc18-1 (as suggested by the label in the figure) but the others included Munc18-1, this should be explicitly explained in the legend.

Following the reviewer’s suggestion, we have separated the FECs in the original Figure 5 into two figures (new Figures 5 and 7) based on the presence and absence of Munc18-1.